ARTICLES

# Gene–environment correlations across geographic regions affect genome-wide association studies

Abdel Abdellaoui[1]✉, Conor V. Dolan[2], Karin J. H. Verweij[1] and Michel G. Nivard[2]

**Gene–environment correlations affect associations between genetic variants and complex traits in genome-wide association studies (GWASs). Here we showed in up to 43,516 British siblings that educational attainment polygenic scores capture gene–environment correlations, and that migration extends these gene–environment correlations beyond the family to broader geographic regions. We then ran GWASs on 56 complex traits in up to 254,387 British individuals. Controlling for geographic regions significantly decreased the heritability for socioeconomic status (SES)-related traits, most strongly for educational attainment and income. For most traits, controlling for regions significantly reduced genetic correlations with educational attainment and income, most significantly for body mass index/body fat, sedentary behavior and substance use, consistent with gene–environment correlations related to regional socio-economic differences. The effects of controlling for birthplace and current address suggest both passive and active sources of gene–environment correlations. Our results show that the geographic clustering of DNA and SES introduces gene–environment correlations that affect GWAS results.**

Genome-wide association studies (GWASs) are an important tool for the investigation of the epidemiology and biology of mental and physical health outcomes. GWASs are viewed as essential herein, because for most outcomes in life, individual differences are a consequence of a multitude of genetic and environmental influences[1]. The aim of a GWAS is to estimate associations between genetic variants and complex traits[2]. The nature of the associations is generally difficult to determine, especially for highly polygenic traits[3]. Genetic effects are expressed through many cascades of biological processes that influence and react to environmental exposures, which can give rise to gene–environment correlations. Three potential sources of gene–environment correlations are as follows: (1) population stratification, (2) family-level gene–environment correlations and (3) gene–environment correlations across geographic regions.

Population stratification affects GWASs when trait differences co-vary with systematic allele frequency differences between ancestries[4,5]. Genome-wide allele frequency differences between populations due to genetic drift and/or natural selection are detectable even within reasonably homogenous populations[6–8]. These older ancestry differences generally show strong correlations with geography[6–9] and could therefore align with regional differences in environmental influences. Population stratification could then lead to biases in GWASs that are not only due to systematic genetic differences but also due to correlated environmental differences between (sub)populations. Bias due to population stratification in GWASs is commonly controlled for with principal components (PCs) that reflect the strongest axes of (ancestral) genetic variation[10].

Differences between families in (socio-economic) environmental factors can also induce gene–environment correlations. Outcomes can, for example, be influenced by the parental genotype through parental (rearing) behavior and inherited socio-economic environments[11]. These indirect genetic effects have been detected for genes associated with educational attainment on a variety of outcomes. Because parental and offspring genotypes correlate, polygenic effects on education could be mixed with signals of GWASs on

physical and mental health outcomes that are influenced by these environmental factors. Some refer to these indirect genetic effects as 'nature of nurture' (ref. [11]), implying indirect effects that arise by ways of parenting or nurturing a child. Others call it 'dynastic effects' (ref. [12]), implying that indirect effects can arise from the succession of (economic) (dis)advantage accumulated across generations, improving children's (socio-economic) positions at birth. As families are nested in neighborhoods, regions and other social structures, accounting for family can partly account for effects at other levels as well.

A third source of gene–environment correlations can be found across geographic regions, which could result from both active and passive processes. Active processes could occur when individuals with favorable genetic predispositions are more likely to improve their environmental circumstances through migration[13]. This process can increase the geographic clustering of genes associated with educational attainment and decrease the geographic clustering of PCs that capture older ancestry differences[13]. Migration can improve rearing environments for offspring as well, making active gene–environment correlations in one generation induce family-level gene–environment correlations in the next. Passive sources of regional gene–environment correlation could arise, for example, due to government policies that affect certain socioeconomic strata more than others. For instance, a policy change that makes insulin more expensive will introduce a correlation between alleles related to socioeconomic status (SES) (for example, education and income) and those related to consequences of untreated diabetes.

In this study, we investigated the effects of gene–environment correlations across geographic regions on polygenic signals for a wide range of traits. We first examined passive and active gene–environment correlations at the family and regional level. To this end, we used educational attainment polygenic scores and phenotypic measures of 56 complex traits in 22,657 adult sibling pairs from UK Biobank. We then conducted GWASs on 56 complex traits in a dataset of up to 254,557 adult individuals of European descent from Great Britain (UK Biobank)[14]. In these GWASs, we reduced the part

[1]Department of Psychiatry, Amsterdam UMC, University of Amsterdam, Amsterdam, The Netherlands. [2]Department of Biological Psychology, VU University, Amsterdam, The Netherlands. ✉e-mail: a.abdellaoui@amsterdamumc.nl

1345

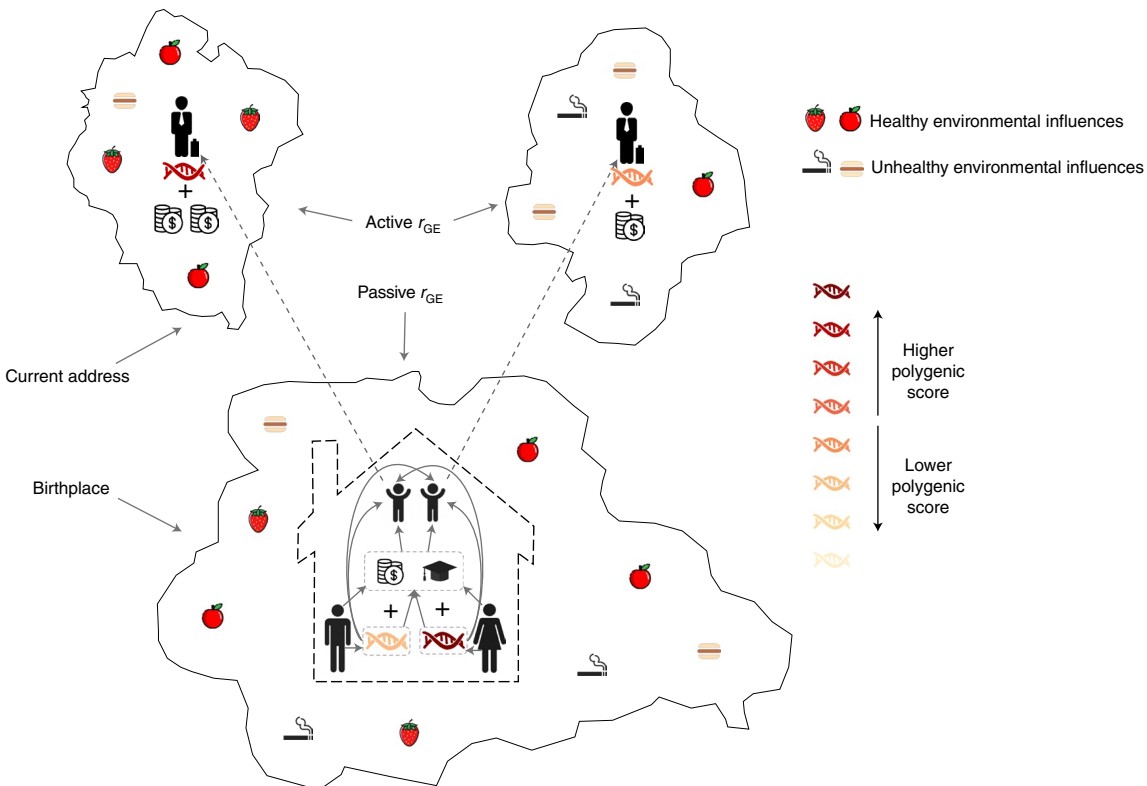

**Fig. 1 | Schematic illustration of gene–environment correlations ($r_{GE}$).** The geographic region at the bottom is the birthplace of the sibling pair, who migrate as adults to two different regions at the top; the sibling with a higher polygenic score migrates to the top-left region with healthier environmental influences. Passive gene–environment correlations occur when the environment that parents provide (at the bottom) correlates with a heritable trait, and active gene–environment correlations come about when heritable behaviors (for example, migration; from bottom to top) lead to correlations between polygenic effects and environmental factors.

of the signal that arises due to gene–environment correlations by introducing fixed effects for neighborhoods that are economically more homogenous than the country as a whole, effectively performing within-region GWASs for a wide range of complex traits. We investigated the impact of gene–environment correlations on the heritability of complex traits and on their genetic relationships with socio-economic outcomes (education and income).

## Results

We first show gene–environment correlations across geographic regions using polygenic scores in siblings. We then reduce their impact on GWAS signals by controlling for geographic regions. For both parts, we analyzed 1,246,531 common SNPs and 56 complex traits related to physical and mental health, body composition, and emotional, cognitive, behavioral and socioeconomic outcomes (see Supplementary Table 1 for a list of traits and sample sizes). In all analyses, we controlled for population stratification (100 PCs), sex and age.

**Polygenic scores in siblings.** In the presence of gene–environment correlations, polygenic scores are predictive of both genetic and environmental influences on the trait. Selzam et al. demonstrated in 2,366 dizygotic twin pairs that, for educational attainment and IQ, polygenic scores explain 60% more variance between families than within families[15]. This study was done in children and young adolescents, aged 12–21 years, where much of the gene–environment correlations originated from the family (shared) environment in which they were born and raised, that is, passive gene–environment correlations. We conducted similar analyses in this study in 22,675 adult sibling pairs aged 40–70 years. We extended our analyses

to investigate whether gene–environment correlations across geographic regions result in an increase in predictive power and to test whether this was influenced by active gene–environment correlations. Active gene–environment correlations could occur, for example, when adults with a higher educational attainment polygenic score are more likely to move to a more favorable region than their siblings with a lower educational attainment polygenic score (Fig. 1).

SES-related signals are best captured by GWASs on educational attainment and income. We created a polygenic score for educational attainment rather than income because educational attainment had a large GWAS excluding UK Biobank participants (245,621 European participants excluding British cohorts)[16]. We used this polygenic score to predict educational attainment as well as 55 other complex traits, as we expected environmental effects involved in gene–environment correlations to impact a wider variety of outcomes. We tested for the presence of gene–environment correlations on the family level and across geographic regions by fitting five models: **model 1** includes polygenic scores on an individual level as a predictor; **model 2** includes polygenic scores on the within-family and between-family level as predictors; **model 3** includes polygenic scores on the within-region and between-region level as predictors; **model 4** includes polygenic scores on the within-family (within-region), between-family (within-region) and between-region level as predictors; in **model 5**, we decomposed the within-family polygenic score into within- and between-region effects to test for active gene–environment correlations. Regions are based on Middle Layer Super Output Area (MSOA) regions of the current address (Fig. 2). An important indicator for the presence of gene–environment correlations beyond the family level across

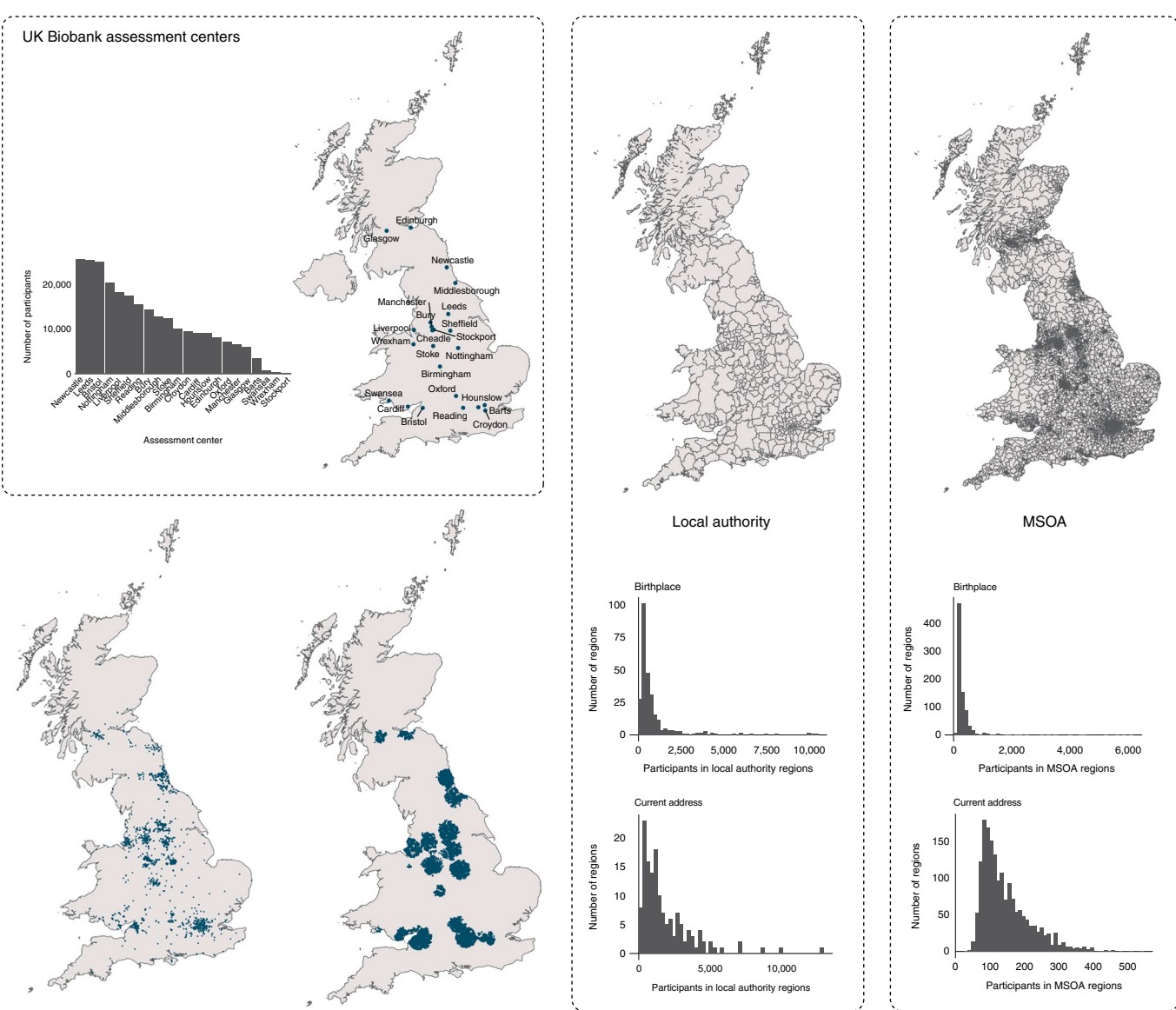

**Fig. 2 | Assessment centers, local authority and MSOA regions.** The two bottom maps show the locations of birthplace and current address of participants analyzed in the GWASs. Histograms show distributions of these participants across assessment centers and geographic regions for birthplaces and current addresses. All maps and histograms show the same 254,577 UK Biobank participants. Maps were adapted from 2011 Census aggregate data (UK Data Service, February 2017 edition). Office for National Statistics; National Records of Scotland; Northern Ireland Statistics and Research Agency (2017).

geographic regions is a weaker within-family effect in model 4 than in model 2, as that would imply that a portion of the effect captured on the individual level, and not by the family, is due to gene–environment correlations across geographic regions. We determined the significance of this difference by bootstrapping (see Methods for more details on the models).

We fitted the five models to the data of 56 complex traits with sample sizes ranging from 11,093 to 43,516 individuals. Main results are shown in Fig. 3 and Extended Data Fig. 1. As expected from ref. [15], individual-level effects became generally weaker after adding between-family effects (model 1 versus model 2; Extended Data Fig. 1a). For ten traits, the individual-level effect of the polygenic score became significantly weaker after adding a between-region effect to a model with a between-family effect (that is, weaker in model 4 compared to model 2, based on the false discovery rate (FDR)-corrected $p$-values of the difference in effect sizes based

on 1,000 bootstraps; Fig. 3a). The five most significant reductions were observed for body mass index (BMI) ($P = 1 \times 10^{-5}$), waist circumference ($P = 1 \times 10^{-4}$), household income ($P = 1 \times 10^{-4}$), time spent watching television ($P = 3 \times 10^{-4}$) and whole-body fat mass ($P = 6 \times 10^{-4}$), possibly because, of all traits considered, these are most likely to change once siblings migrate away from the parental residence (see Table 1 for full results for these traits). Adding the between-region effect to the between-family effect also significantly decreased the individual-level polygenic score effect for educational attainment ($P = 9 \times 10^{-4}$). Six traits showed an increase in $R^2 > 2\%$ after adding the between-region effect to the between-family effect (Fig. 3b). When comparing fixed effect estimates within model 4, the majority of traits showed higher between-region than between-family effect (Fig. 3c), implying additional explanatory power of geographic regions not captured at the individual or family level. Part of the between-family effect in model 2 can be explained

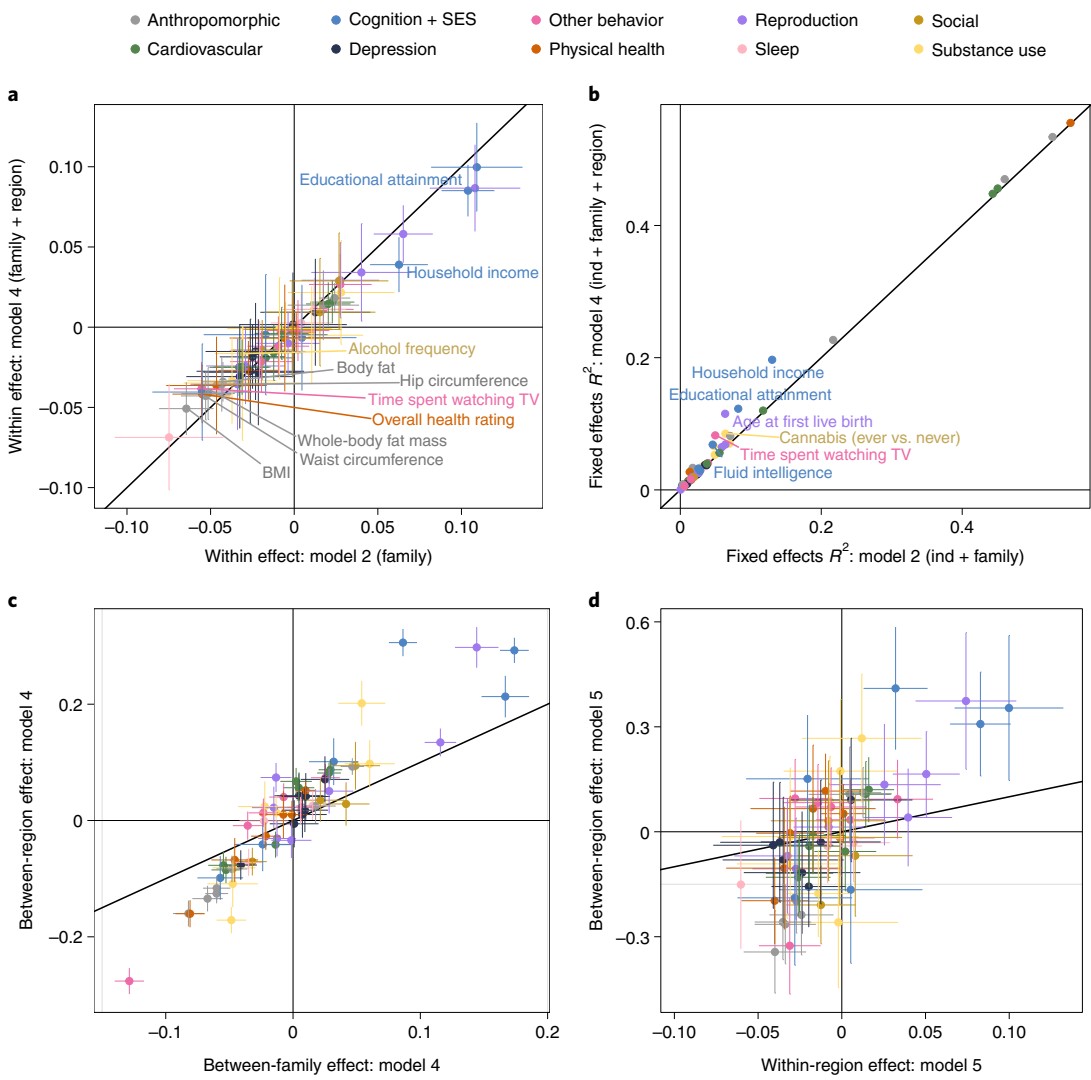

**Fig. 3 | Results of sibling educational attainment polygenic score analyses for 56 complex traits.** Sample sizes range from 11,093 to 43,516 (see Supplementary Table 1 for sample size per trait). Error bars in **a**, **c** and **d** indicate 95% confidence intervals. **a**, Comparison of the within-effect estimate of model 2 and the within-effect estimate of model 4. Trait names are shown for the ten traits that showed a significant decrease (*p*-value of the difference based on 1,000 bootstraps). **b**, Comparison of the marginal $R^2$ of models 2 and 4 (that is, variance explained by all fixed effects, including age and sex). Trait names are shown for the six traits for which the difference between models 2 and 4 was more than 2%. **c**, Comparison of the between-family effect and between-region effect estimates of model 4. **d**, Comparison of the within-region effect and between-region effect estimates of model 5.

by between-region effects, as reflected by a significant decrease of the between-family effect in the majority of traits after adding the between-region effect (model 4 versus model 2; Extended Data Fig. 1b). After decomposing the within-family effect into within- and between-region effects (model 5), the majority of traits showed higher between-region than within-region effect (Fig. 3d), implying wide-spread active gene–environment correlations.

**GWASs controlled for geography.** We ran linear mixed model (LMM)[17] GWASs on 56 complex traits in up to 254,387 participants of European descent. We included the geographic region of birth and/or current residence of the participants as dummy coded fixed effects to control for gene–environment correlations. Geographic regions were obtained by mapping latitude and longitude coordinates of birthplace or current address (1-km resolution) to local authority areas or MSOA regions (Fig. 1). The local authority areas are based on the level of subnational division used for the purposes of local government, while MSOA regions are defined as a set of adjacent output areas chosen to have comparable population sizes

and to be "as socially homogenous as possible based on tenure of household and dwelling type" (ref. [18]). Both local authority regions and MSOA regions explained significant amounts of variation for all 56 complex traits, most for household income (~5% based on birthplace, ~14% based on current address; Extended Data Fig. 2). Current address consistently explained more variation than birthplace, in line with richer and healthier people being more likely to have moved to regions inhabited by richer and healthier people[13].

We only included regions with ≥ 100 UK Biobank participants and selected only participants with all geographic information available. For local authority areas, this resulted in 282 regions for birthplace and 147 regions for current address. For MSOA regions, this resulted in 816 regions for birthplace and 1,747 regions for current address. A total of 254,557 participants with geographic regions were available for both birthplace and current address (Fig. 2). Of these participants, between 63,780 and 254,387 had phenotypes available (Supplementary Table 1). In addition, we compared the results with those obtained when controlling the GWASs for latitude and longitude coordinates (also for birthplace, current address

**Table 1 | Results from models 1–5 for the top five traits that showed significant evidence of gene–environment correlations that extended beyond the family level across geographic regions**

| | 1. Individual-level polygenic score only | | 2. Within + between family | | 3. Within + between region | | 4. Within + between family + between region | | 5. Within family fixed effect split into within and between region fixed effect | |
|---|---|---|---|---|---|---|---|---|---|---|
| | Estimate (SE) | t-value (p) | Estimate (SE) | t-value (p) | Estimate (SE) | t-value (p) | Estimate (SE) | t-value (p) | Estimate (SE) | t-value (p) |
| **BMI** | | | | | | | | | | |
| Within/individual level | −0.10 (0.005) | −19.29 ($2 \times 10^{-82}$) | −0.06 (0.008) | −7.77 ($8 \times 10^{-15}$) | −0.07 (0.005) | −14.23 ($7 \times 10^{-46}$) | −0.05 (0.008) | −6.04 ($1 \times 10^{-9}$) | −0.04 (0.009) | −4.21 ($3 \times 10^{-5}$) |
| Between family | – | – | −0.11 (0.006) | −18.22 ($1 \times 10^{-73}$) | – | – | −0.08 (0.006) | −13.26 ($6 \times 10^{-40}$) | – | – |
| Between region | – | – | – | – | −0.20 (0.007) | −29.75 ($1 \times 10^{-181}$) | −0.16 (0.011) | −14.10 ($6 \times 10^{-45}$) | −0.34 (0.060) | −5.73 ($1 \times 10^{-8}$) |
| Marginal R² (fixed effects only) | 0.016 | | 0.018 | | 0.033 | | 0.033 | | 0.009 | |
| Conditional R² (fixed and random effects) | 0.291 | | 0.292 | | 0.300 | | 0.300 | | 0.302 | |
| **Waist circumference** | | | | | | | | | | |
| Within/individual level | −0.07 (0.004) | −16.55 ($3 \times 10^{-6}$) | −0.05 (0.008) | −7.02 ($2 \times 10^{-12}$) | −0.05 (0.004) | −12.06 ($2 \times 10^{-33}$) | −0.04 (0.008) | −5.61 ($2 \times 10^{-8}$) | −0.04 (0.009) | −4.10 ($4 \times 10^{-5}$) |
| Between family | – | – | −0.08 (0.005) | −15.43 ($2 \times 10^{-53}$) | – | – | −0.06 (0.005) | −1.95 ($8 \times 10^{-28}$) | – | – |
| Between region | – | – | – | – | −0.15 (0.006) | −24.56 ($3 \times 10^{-127}$) | −0.12 (.010) | −11.25 ($3 \times 10^{-29}$) | −0.26 (0.055) | −4.70 ($3 \times 10^{-6}$) |
| Marginal R² (fixed effects only) | 0.217 | | 0.217 | | 0.227 | | 0.227 | | 0.216 | |
| Conditional R² (fixed and random effects) | 0.411 | | 0.411 | | 0.424 | | 0.424 | | 0.428 | |
| **Household income** | | | | | | | | | | |
| Within/individual level | 0.12 (0.005) | 25.10 ($8 \times 10^{-138}$) | 0.06 (0.01) | 7.12 ($1 \times 10^{-12}$) | 0.07 (0.005) | 15.25 ($2 \times 10^{-52}$) | 0.04 (0.01) | 4.46 ($8 \times 10^{-6}$) | 0.03 (0.01) | 3.30 ($1 \times 10^{-3}$) |
| Between family | – | – | 0.15 (0.01) | 25.47 ($6 \times 10^{-141}$) | – | – | 0.09 (0.01) | 15.35 ($6 \times 10^{-53}$) | – | – |
| Between region | – | – | – | – | 0.33 (0.01) | 44.39 ($<10^{-310}$) | 0.31 (0.01) | 25.69 ($3 \times 10^{-143}$) | 0.41 (0.09) | 4.58 ($5 \times 10^{-6}$) |
| Marginal R² (fixed effects only) | 0.127 | | 0.130 | | 0.196 | | 0.197 | | 0.127 | |
| Conditional R² (fixed and random effects) | 0.320 | | 0.321 | | 0.367 | | 0.368 | | 0.377 | |
| **Time spent watching television** | | | | | | | | | | |
| Within/individual level | −0.14 (0.005) | −27.74 ($1 \times 10^{-167}$) | −0.06 (0.008) | −6.60 ($4 \times 10^{-11}$) | −0.10 (0.005) | −2.74 ($5 \times 10^{-95}$) | −0.04 (0.009) | −4.51 ($6 \times 10^{-6}$) | −0.03 (0.009) | −3.32 ($9 \times 10^{-4}$) |
| Between family | – | – | −0.17 (0.006) | −29.07 ($3 \times 10^{-182}$) | – | – | −0.13 (0.006) | −21.95 ($1 \times 10^{-105}$) | – | – |
| Between region | – | – | – | – | −0.29 (0.007) | −42.51 ($<10^{-310}$) | −0.28 (0.011) | −24.24 ($4 \times 10^{-128}$) | −0.33 (0.070) | −4.62 ($4 \times 10^{-6}$) |
| Marginal R² (fixed effects only) | 0.044 | | 0.050 | | 0.080 | | 0.082 | | 0.029 | |
| Conditional R² (fixed and random effects) | 0.267 | | 0.273 | | 0.279 | | 0.284 | | 0.299 | |
| **Whole-body fat mass** | | | | | | | | | | |
| Within/individual level | −0.08 (0.005) | −16.20 ($8 \times 10^{-59}$) | −0.05 (0.008) | −6.32 ($3 \times 10^{-10}$) | −0.06 (0.005) | −11.85 ($3 \times 10^{-32}$) | −0.04 (0.008) | −5.02 ($5 \times 10^{-7}$) | −0.03 (0.009) | −3.62 ($3 \times 10^{-4}$) |
| Between family | – | – | −0.09 (0.006) | −15.44 ($2 \times 10^{-53}$) | – | – | −0.07 (0.006) | −11.06 ($2 \times 10^{-28}$) | – | – |
| Between region | – | – | – | – | −0.17 (0.007) | −25.14 ($8 \times 10^{-133}$) | −0.13 (0.011) | −12.02 ($3 \times 10^{-33}$) | −0.26 (0.058) | −4.60 ($5 \times 10^{-6}$) |
| Marginal R² (fixed effects only) | 0.070 | | 0.071 | | 0.081 | | 0.082 | | 0.063 | |
| Conditional R² (fixed and random effects) | 0.328 | | 0.327 | | 0.339 | | 0.338 | | 0.338 | |

This table contains results from linear mixed effect models, and reports the fixed effect estimates, their standard errors (SE), and the t-value and corresponding uncorrected one-sided P-value of the significance of the fixed effect.

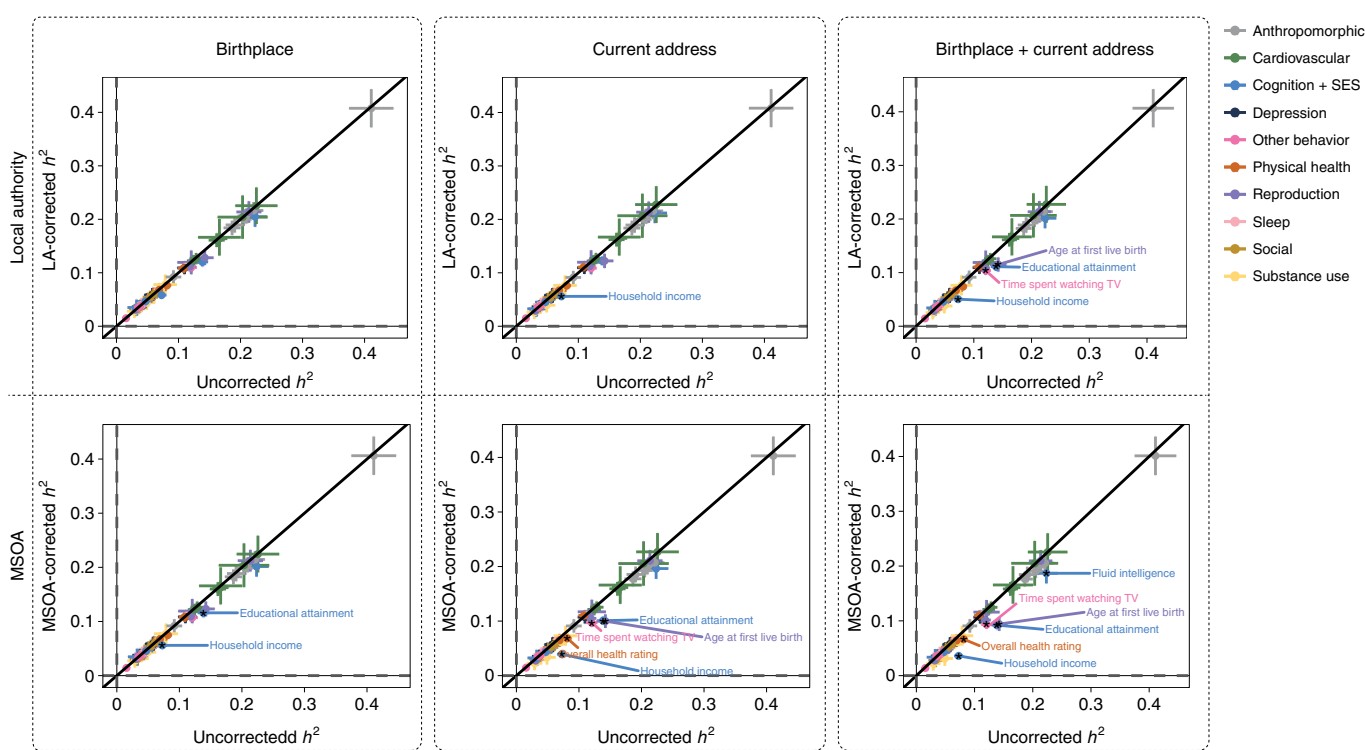

**Fig. 4 | SNP-based heritabilities of 56 complex traits, corrected and uncorrected for geographic region.** The panels show the estimated SNP-based heritabilities after controlling for local authority or MSOA regions based on birthplace and/or current address. Error bars indicate 95% confidence intervals. Trait names are shown for traits with a significant change in SNP-based heritability (FDR-corrected $P < 0.05$). Sample sizes of the GWASs ranged from 63,780 to 254,387 (see Supplementary Table 1 for sample size per trait).

and both) and assessment center. As we explain in the Methods, the current address measure is likely to be more precise than the birthplace measure. We compared GWAS results controlling for region with the results of conventional GWASs not corrected for region, obtained from the same selection of individuals. The impact of controlling for geographic region was investigated by computing the magnitude and significance of the change in SNP-based heritability and the change in genetic correlation with two indicators of SES, namely educational attainment and household income.

*SNP-based heritability.* Without controlling for geographic region, the SNP-based heritabilities of the 56 traits ranged from 0.02 to 0.41 (estimated using linkage disequilibrium (LD) score regression[19] based on ~1.2 million SNPs), with physical traits showing higher heritability estimates than behavioral traits (Fig. 4). The average SNP-based heritability per category was as follows: anthropomorphic, 0.22; cardiovascular, 0.17; cognition and SES, 0.10; depression, 0.06; physical health, 0.07; reproduction, 0.10; sleep, 0.07; social, 0.05; substance use, 0.05 and other behavior, 0.05. Figures 4 and 5 show the change in SNP-based heritability after controlling for geographic region. All heritability estimates before and after controlling for geographic region can be found in the Supplementary Data file.

The heritability estimates most consistently affected by controlling for local authority or MSOA regions were those of educational attainment and income. Controlling for current address resulted in stronger heritability reductions than controlling for birthplace, and including both resulted in the strongest reductions. Controlling for the smaller MSOA regions resulted in stronger heritability reductions than controlling for the larger local authority regions. Accordingly, controlling for MSOA regions based on both birthplace and current address resulted in significant heritability reductions for the largest number of traits ($n = 6$): household income

(from 7% to 4%), educational attainment (from 14% to 9%), age at first birth (from 14% to 9%), time spent watching television (from 12% to 9%), overall health (from 8% to 7%) and fluid intelligence (from 22% to 19%). These traits have a strong relationship with SES, especially income and educational attainment, which showed the strongest reduction and also the strongest geographic clustering (Extended Data Fig. 2), making it likely that their polygenic signal was larger because it captured effects attributable to gene–environment correlations.

Assessment center did not significantly impact any of the heritability estimates, while latitude and longitude did, but for very different traits, namely anthropomorphic and cardiovascular traits (Extended Data Figs. 3 and 4). The SNP-based heritability of height was most strongly affected by latitude and longitude corrections, decreasing from ~41% to ~13% (for both birthplace and current address separately and combined). Other significant reductions were observed for hip circumference, corpuscular volume and corpuscular hemoglobin. These effects could not be explained by a reduction in population stratification, as the LD score intercept increased after controlling for latitude and longitude (Supplementary Table 2). This heritability reduction only occurred when controlling for 100 PCs and latitude and longitude combined but not when controlling for only PCs or only longitude and latitude (Supplementary Table 2); because latitude and longitude correlate strongly with several PCs (Supplementary Table 3), this effect may be caused by multicollinearity.

*Genetic correlations.* Genes associated with socioeconomic success can influence which neighborhoods people can afford to live in, and thus the quality of people's living environment. Environmental exposures that differ between neighborhoods and regions can affect a wide range of physical and mental health outcomes, which causes

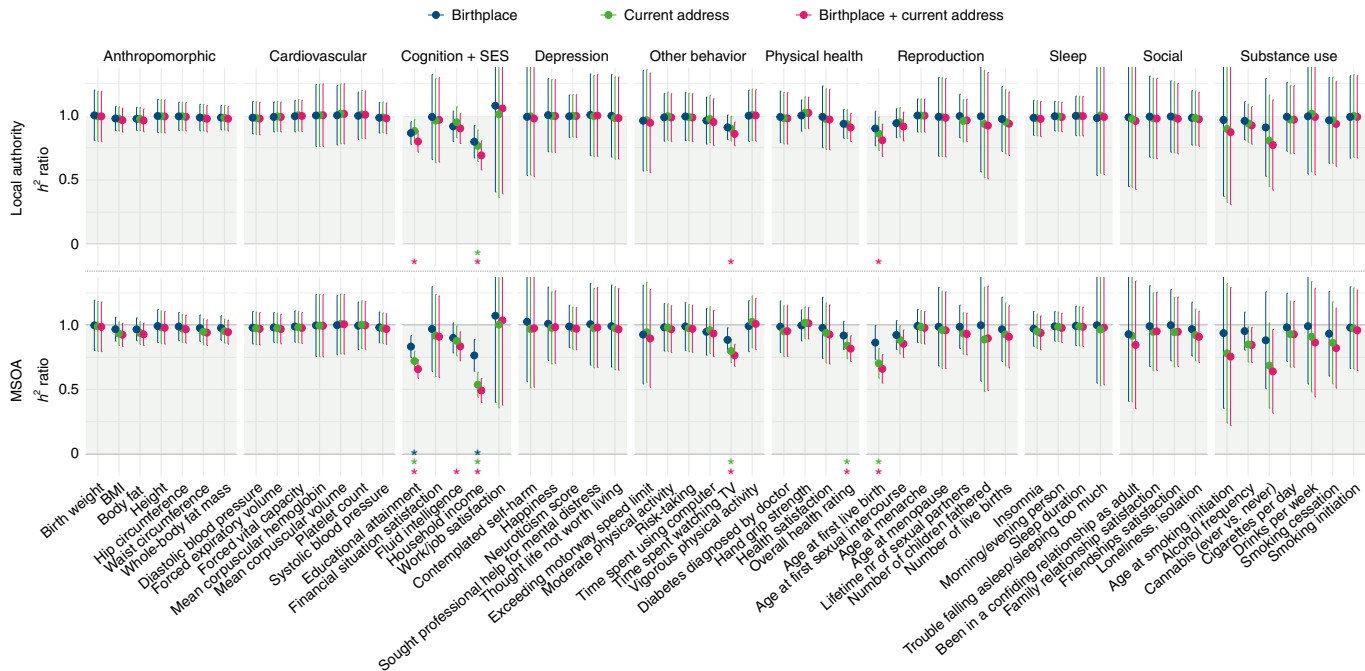

**Fig. 5 | The ratio of decrease in SNP-based heritability after controlling for geographic region for 56 complex traits.** The ratio of the decrease was computed by dividing the corrected SNP-based heritability estimate by the uncorrected SNP-based heritability estimate. Error bars indicate 95% confidence intervals. Asterisks indicate FDR-corrected $P < 0.05$, corresponding to a significant change in SNP-based heritability. Sample sizes of the GWASs ranged from 63,780 to 254,387 (see Supplementary Table 1 for sample size per trait).

genes associated with socioeconomic success to also become associated with these physical and mental health outcomes. We investigated whether controlling for regional differences decreases genetic correlations with SES. Genetic correlations were computed for 56 complex traits with educational attainment and household income using LD score regression[20] and ~1.2 million SNPs. The significance of the change in genetic correlations after controlling for geography was tested in Genomic SEM[21] accounting for the dependence between the various GWASs, which all rely on the same sample. Figure 6 and Extended Data Fig. 5 show the changes in genetic correlations and their significance. All uncorrected and corrected genetic correlations with educational attainment and household income can be found in Extended Data Figs. 6 and 7 and in the Supplementary Data file.

Controlling for local authority or MSOA region significantly reduced the genetic correlations between most of the 56 complex traits and education and income (Fig. 6). We observed the most and strongest reductions when controlling for birthplace and current address jointly and the smaller MSOA regions. When controlling for MSOA regions based on both birthplace and current address jointly, 41 traits showed a significant reduction in the genetic correlation with educational attainment. The five most significant decreases were observed for height (from 0.20 to 0.11, $P_{change} = 9 \times 10^{-70}$), body fat percentage (from −0.34 to −0.20, $P_{change} = 1 \times 10^{-69}$), BMI (from −0.31 to −0.17, $P_{change} = 1 \times 10^{-68}$), alcohol frequency (from −0.42 to −0.23, $P_{change} = 1 \times 10^{-65}$) and time spent watching television (from −0.69 to −0.54, $P_{change} = 2 \times 10^{-63}$). When controlling for MSOA regions based on birthplace and current address, 35 traits showed a significant reduction of the genetic correlation with household income. The five most significant decreases were observed for body fat percentage (from −0.32 to −0.15, $P_{change} = 5 \times 10^{-44}$), BMI (from −0.33 to −0.15, $P_{change} = 7 \times 10^{-41}$), time spent watching television (from −0.62 to −0.41, $P_{change} = 2 \times 10^{-37}$), whole-body fat mass (from −0.25 to −0.09, $P_{change} = 3 \times 10^{-37}$) and waist circumference (from

−0.29 to −0.13, $P_{change} = 5 \times 10^{-34}$), which is the same top five as for the polygenic score analyses in siblings summarized in Table 1.

A relatively small number of traits showed significantly stronger genetic correlations with educational attainment after controlling for geographic region. When controlling for MSOA region based on both birthplace and current address, genetic correlations with education became significantly stronger for eight traits (number of sexual partners, corpuscular volume, corpuscular hemoglobin, risk-taking, age at menarche, job satisfaction, drinks per week and smoking initiation) and genetic correlations with income became significantly stronger for six traits (ever sought/received professional help for mental distress, number of sexual partners, corpuscular volume, corpuscular hemoglobin, happiness and family relationship satisfaction). This could mean that regional differences in SES masked genetic correlations between these traits and educational attainment, but could potentially also result from collider bias (Discussion and Methods).

We also computed genetic correlations between cognitive and noncognitive[22] components of educational attainment (Extended Data Figs. 7 and 8). Similar changes occurred after controlling for MSOA region, except for a small number of traits differentially correlated with cognitive versus noncognitive skills, which, accordingly, showed differential effects for controlling for geography. Interestingly, income showed a particularly strong difference, with little change in cognitive skills after controlling for MSOA, but a significant decrease in its genetic correlation with noncognitive skills.

## Discussion
Environmental factors that differ between economically prosperous and deprived regions influence a wide range of complex traits, while these regions can also show genetic differences. Within-family designs can reduce the impact of these gene–environment correlations on GWASs[23–25], but they do not control for active gene–environment correlations that occur when genetic differences

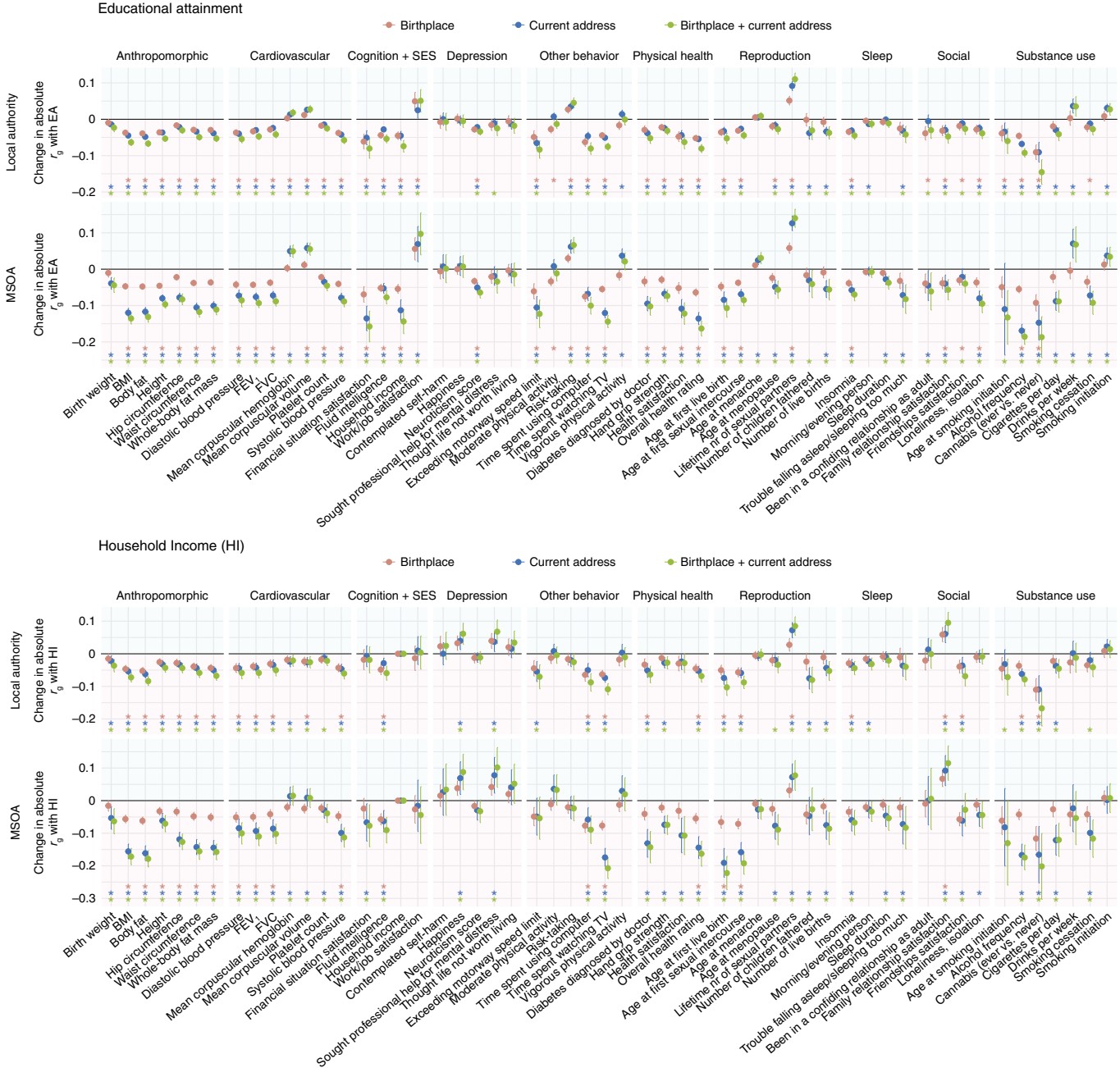

**Fig. 6 | The change in absolute genetic correlations with educational attainment (EA, top) and household income (HI, bottom).** The genetic correlations were computed with LD score regression. The change in absolute genetic correlation is shown to visualize the change in the strength of the genetic relationships with educational attainment or household income (the directions of the genetic correlations vary between traits and are displayed in Extended Data Figs. 3 and 4). Error bars indicate 95% confidence intervals. Asterisks indicate FDR-corrected $P < 0.05$, corresponding to a significant change in genetic correlation. Sample sizes of the GWASs ranged from 63,780 to 254,387 (see Supplementary Table 1 for sample size per trait).

between family members become associated with SES-related neighborhood characteristics after children leave the parental residence. Furthermore, large sample sizes of genotyped families are harder to attain, as reflected in noisier genetic effect estimates of current within-family GWASs (Extended Data Fig. 9). We showed here that the effects of gene–environment correlations that extend beyond the family can be reduced in GWASs by controlling for geographic regions. Fully removing polygenic effects that affect the outcome via the social environment will be more challenging, and it depends on the research question of the GWAS whether it would be desirable to exclude this causal chain from the signal[26].

We confirmed the presence of gene–environment correlations across geographic regions by analyzing associations between educational attainment polygenic scores and 56 complex traits in adult siblings. A significant part of the predictive power of the polygenic score could be attributed to gene–environment correlations across families and across geographic regions. Geographic regions capture a substantial part of family-level gene–environment correlation and a significant, but smaller, portion of independent gene–environment correlations across regions. These additional gene–environment correlations were less significant for educational attainment than for traits, such as BMI, income and time spent watching

television, even though these analyses were done with an educational attainment polygenic score. This may be explained by people obtaining their education before migrating to their current address, in which case passive gene–environment correlations would have a stronger impact than active gene–environment correlations on educational attainment.

Controlling GWASs for geographic region significantly reduced SNP-based heritability estimates for a relatively small number of SES-related outcomes, most significantly for income, educational attainment, age at first birth, time spent watching television, overall health and fluid intelligence. Genetic correlations with SES (education and income) decreased significantly for most traits, more so when controlling for current address than for birthplace, in line with increased geographic clustering of complex trait variation (and decreased geographic clustering of ancestry-related variation)[13] due to SES-related migration. Parents' addresses are likely influenced by the same genetic variants inherited by their offspring, and there are multiple mechanisms that could lead to offspring increasing the already significant gene–environment correlation through their own migration in adulthood. These mechanisms could be environmental and/or genetic, including wealth transfer from parent to child, or offspring potentially having higher polygenic scores than their parents due to assortative mating, which is especially strong for SES[27,28].

The most significant reductions in genetic correlations with education and income were observed for traits related to BMI and body fat, suggesting correlations between SES-related genes and obesogenic environments. This may help (partly) explain why the polygenic score for BMI shows the strongest geographic clustering in Great Britain, after educational attainment and cognition[13]. While the relationship between BMI and SES is positive in poorer countries, where food insecurity relates to food shortages, it is reversed in more developed countries, where food insecurity relates to a lack of access to healthy foods[29]. The number of fast-food restaurants and diabetes rates in British neighborhoods are significantly correlated with regional differences in (polygenic scores for) educational attainment[13]. Behavioral traits related to body weight, namely time spent watching television and frequency of alcohol intake, are also among the top five traits showing these gene–environment correlations in their polygenic signals. This suggests that, besides dietary options, behaviors that correlate with regional socio-economic factors may also affect regional differences in body weight and their associated polygenic signals.

There are several limitations to be kept in mind when interpreting our results. Firstly, UK Biobank participants are on average more highly educated and healthier than the general population, and they are more often from regions that are better developed economically[30,31]. Geographic clustering of SES is probably stronger in the general population than in UK Biobank[13], so effects observed in the current study may be underestimated. Furthermore, the geographic coverage is wider for birthplace than for current address, as participants were sampled to live within a radius of 35 km of one of the 22 UK Biobank assessment centers. Controlling for assessment center, however, had minimal impact on the GWAS signal compared to the other regional measures (Extended Data Figs. 3–5). The geographic locations that we use are not precise; to ensure anonymity, birthplace and current address have been rounded to 1 km and the current address location is more precise than birthplace (Methods). While boundaries of MSOA regions were chosen to delineate socially homogeneous regions[18], the geographic location of participants is only a relatively crude and temporally variable proxy for social environments that underlie the gene–environment correlations that we try to control for. Environmental circumstances start with the family environment and expand to close and extended social circles, likely across multiple communities throughout life. It will be challenging to fully understand and account for correlations between genes and environments, but there is room for improvement by genotyping family members, social circles and collecting longitudinal information on the participants' living environments. Furthermore, when correcting for geographic location, there is a risk of collider bias: if genetic variants affect current address (for example, through effects on cognitive ability) and the outcome of interest (for example, substance use) also affects current address, controlling for current address in the GWAS on substance use may induce collider bias, biasing estimates of SNP associations, heritability and genetic correlations (Methods and Extended Data Fig. 10). Finally, controlling for current address only gets us closer to the direct genetic effect on the outcome if the relationship between environment and outcome arises from a causal effect of the current environment on the outcome. The outcomes for which we detect the strongest evidence for active gene–environment correlations (for example, BMI, time spent watching television) are plausible outcomes of current adult environments, while for other outcomes it is more plausible that they contributed to the current environment (for example, educational attainment).

Effects estimated in GWASs of many phenotypes are affected by gene–environment correlations, and these are not entirely attributable to processes that take place within a family but are also attributable more broadly to regional social, economic and political processes that correlate with individuals' genotypes. We showed how to significantly reduce these effects in GWAS signals. It will depend on the goals of the research whether this would be desirable, as these effects result from dynamic social circumstances that are part of a true causal chain in between our DNA and complex mental and physical health outcomes. Statistical models, research designs and conclusions of GWASs need to more carefully reflect the reality of the social and geographic structure of society.

## Online content

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

## Methods

**Participants.** The participants of this study come from UK Biobank[14,32]. UK Biobank has received ethical approval from the National Health Service North West Centre for Research Ethics Committee (reference: 11/NW/0382). A total of 273,402 females and 229,134 males ($N_{total} = 502,536$) aged between 37 and 73 years were recruited between 2006 and 2010 across 22 assessment centers throughout Great Britain. The sampling strategy was aimed to cover a variety of different settings providing socio-economic and ethnic heterogeneity and an urban–rural mix. The participants underwent cognitive, health and lifestyle assessments, provided blood, urine and saliva samples, and have their health followed longitudinally.

**Genotypes and quality control (QC).** A total of 488,377 participants had their genome-wide SNPs genotyped on UK BiLEVE array ($n = 49,950$) or UK Biobank Axiom Array ($n = 438,423$). Genotypes were imputed using the Haplotype Reference Consortium (HRC) panel as a reference set (pre-imputation QC and imputation are described in more detail in ref. [14]). We extracted SNPs from HapMap3 (1,345,801 SNPs) from the imputed dataset. In the pre-principal-component analysis (PCA) QC on unrelated individuals, we filtered out SNPs with a minor allele frequency (MAF) < 0.01 and missingness > 0.05, leaving 1,252,123 SNPs. After filtering out individuals with non-European ancestry (see 'Ancestry and PCA'), we repeated the SNP QC on unrelated Europeans ($n = 312,927$), filtering out SNPs with MAF < 0.01, missingness > 0.05 and Hardy–Weinberg equilibrium $P < 10 \times^{-10}$, leaving 1,246,531 SNPs. We then created a dataset of 1,246,531 QC-ed SNPs for 456,064 UK Biobank participants of European ancestry. These QC steps were performed using PLINK 2.0.

**Ancestry and PCA.** Ancestry was estimated using PCA. We first determined which participants had non-European ancestry by projecting UK Biobank participants onto the first two PCs from the 2,504 participants of the 1000 Genomes Project, using HapMap3 SNPs with MAF > 0.01 in both datasets. Participants from UK Biobank were assigned to one of five super-populations from the 1000 Genomes project: European, African, East Asian, South Asian, or admixed. Assignments for European, African, East Asian and South Asian ancestries were based on > 0.9 posterior probability of belonging to the 1000 Genomes reference cluster, with the remaining participants classified as admixed. Posterior probabilities were calculated under a bivariate Gaussian distribution, where this approach generalizes the $k$-means method to take account of the shape of the reference cluster. We used a uniform prior and calculated the vectors of means and $2 \times 2$ variance–covariance matrices for each super-population. A total of 456,064 individuals were identified to have European ancestry. To capture ancestry differences within the British population, a PCA was then conducted on these 456,064 individuals of European ancestry. When trying to capture ancestry differences in homogenous populations, genotypes should be pruned for LD with long-range LD regions removed[6]. The LD pruned ($r^2 < 0.2$) UK Biobank dataset without long-range LD regions consisted of 131,426 genotyped SNPs. The PCA to construct British ancestry-informative PCs was conducted on this SNP set for unrelated individuals using flashPCA v2 (ref. [33]). PC SNP loadings were used to project the complete set of European individuals onto the PCs. PCs that reflect ancestry differences are expected to cluster geographically, which we investigated by computing their Moran's I: out of the top 100 PCs, 78 PCs showed significant geographic clustering after Bonferroni correction and 94 PCs showed a $P$-value of < 0.05 (see ref. [13] for details on evaluating the geographic clustering of the PCs using Moran's I).

**Genetic relatedness matrix.** We created genetic relatedness matrices (GRMs) to include in our LMM GWASs to control for cryptic relatedness. The GRMs contain genetic relationships between all individuals based on a slightly LD-pruned HapMap 3 SNP set (LD-pruning parameters used in PLINK: window size = 1,000 variant count, step size = 100, $r^2$ cutoff = 0.9 and MAF > 0.01, resulting in 575,293 SNPs). The GRMs were computed using GCTA[34] on individuals of European descent. We created a sparse GRM, containing only the relationships of related individuals (cutoff = 0.05, resulting in 179,609 relationships) to control for the presence of closely related individuals (cousins, siblings, parent–offspring pairs).

**Polygenic scores.** We computed polygenic scores, that is, the genome-wide sum of alleles weighted by their estimated effect sizes, for educational attainment. The allelic effect size estimates were obtained from the GWAS from ref. [35], which was re-computed to exclude not only UK Biobank dataset (to avoid over-estimation of the genetic predisposition of a trait)[36], but also all British cohorts, as an extra measure against residual population stratification within Great Britain. The polygenic scores were computed with the SBLUP approach[28], which maximizes the predictive power by creating scores with best linear unbiased predictor (BLUP) properties and accounts for LD between SNPs. As a reference sample for the LD, we used a random sample of 10,000 unrelated individuals from UK Biobank that were imputed using the HRC reference panel[37]. To control for population stratification, we regressed out the first 100 PCs from the polygenic scores.

**Phenotypic and geographic measures.** *Phenotypes.* Selection of phenotypic outcomes was based on the relevance of the metric to broad mental and physical health outcomes, resulting in the selection of 56 complex traits encompassing domains of anthropomorphic traits, cardiovascular outcomes, cognition, SES, depressive symptoms, sedentary behavior, reproductive behavior, risk-taking behavior, physical activity, self-reported overall health, sleep, social connection and substance use (Supplementary Table 1). All traits were analyzed as provided by UK Biobank, except for the substance use phenotypes, which were defined according to the GWAS & Sequencing Consortium of Alcohol and Nicotine (GSCAN) in ref. [38], and educational attainment, which was transformed to years of education as defined according to the International Standard Classification of Education (ISCED) coding as analyzed in ref. [16].

*Geographic measures.* Birthplace location was based on the coordinates in UK Biobank fields 129 (latitude) and 130 (longitude). UK Biobank verbal interview includes a procedure to ascertain a participant's birthplace described as follows: "The interviewer is provided with a tree structure that lists place names and counties in England, Wales and Scotland. They were instructed to enter at least the first three letters of the town/village/place that the participant provides. If there are too many matches to the first three letters, no place names appear, and more letters need to be entered. If there is more than one listing of the relevant place name, then they were asked to choose the one with correct district or county. In order to narrow down the search, the interviewer may also need to type in the district or county to find a match. If they cannot find the place name, they were instructed to use the Enter Other facility to enter free text to state the town, county (or district) and country. If the participant does not know, there is the option to enter Unknown. Place of birth in the United Kingdom is then converted into North and East coordinates." It is not clear whether birthplace refers to the residence at birth or the place of the hospital of birth. The current address location was based on the coordinates in UK Biobank fields 22702 (longitude) and 22703 (latitude). Deriving these coordinates is described by UK Biobank as follows: "Where the full address is present and is verified to be valid, software package DataPlus (provided in the QuickAddress Batch packet) is used to transform the address data into the grid coordinates. Where only the postcode is present, the grid coordinates are generated with the aid of online mapping tools: doogal (https://www.doogal.co.uk/) and uk-postcodes (https://www.uk-postcodes.com). Further details of this process are available from relevant websites." The participants' coordinates, which were rounded to 1 km, were mapped to the nearest MSOA region using a shapefile obtained from the InFuse website, which is part of UK Data Service Census Support[39]. The R-packages sp (v1.4-4) and rgdal (v1.5-18) were used to merge the spatial data from the MSOA shapefile[40–42].

**Sibling polygenic score analyses.** We extended the model from ref. [15] using the polygenic score in 22,675 siblings based on an educational attainment GWAS in 245,621 European participants that excluded all British cohorts[16], regressed the first 100 PCs out of the polygenic score to control for population stratification and tested for the presence of gene–environment correlations on the family level as well as across geographic regions by fitting the following four models (covariates of sex and age are included but not shown in the following equations):

Model 1: The model, with the polygenic score on an individual level as fixed effect:

$$Y_{ij} = \alpha_0 + \beta \text{PRS}_{ij} + \gamma_j + \varepsilon_{ij}$$

Model 2: The model with between-family fixed effects added, as used in ref. [15]:

$$Y_{ij} = \alpha_0 + \beta_W \left( \text{PRS}_{ij} - \overline{\text{PRS}_j} \right) + \beta_B \overline{\text{PRS}_j} + \gamma_j + \varepsilon_{ij}$$

Model 3: The model with between-region fixed effects added:

$$Y_{ijk} = \alpha_0 + \beta_W \left( \text{PRS}_{ijk} - \overline{\text{PRS}_k} \right) + \beta_B \overline{\text{PRS}_k} + \gamma_j + \gamma_k + \varepsilon_{ij}$$

Model 4: The model with both between-family and between-region fixed effects added:

$$Y_{ijk} = \alpha_0 + \beta_W \left( \text{PRS}_{ijk} - \overline{\text{PRS}_j} - \overline{\text{PRS}_k} \right) + \beta_{BF} \left( \overline{\text{PRS}_j} - \overline{\text{PRS}_k} \right)$$
$$+ \beta_{BR} \left( \overline{\text{PRS}_k} \right) + \gamma_j + \gamma_k + \varepsilon_{ijk}$$

Here $Y_{ijk}$ denotes the phenotypic outcome, where $i = [1,2]$ corresponds to the sibling (1 is the oldest sibling) from family $j$, who lives in MSOA region $k$. PRS denotes the polygenic score, $\overline{\text{PRS}_j}$ refers to the mean polygenic score of the family and $\overline{\text{PRS}_k}$ refers to the mean polygenic score of the region. $\alpha_0$ denotes the intercept; $\gamma_j$ denotes the random intercept with $\gamma_j \sim \text{N}(0, \sigma^2_{\gamma_j})$, accounting for between-family variation in the intercept; $\gamma_k$ denotes the random intercept with $\gamma_k \sim \text{N}(0, \sigma^2_{\gamma_k})$, accounting for between-region variation in the intercept. The residual is represented as $\varepsilon_{ijk}$ with $\varepsilon_{ijk} \sim \text{N}(0, \sigma^2_{\varepsilon})$. This random intercept mixed-effects model can include multiple fixed effects to separate the association between the polygenic score and the phenotype into within-family (that is, individual-level) and between-family and between-region effects. The random effect term $\sigma^2_{\gamma}$, which estimates the difference between the group intercepts $\gamma_j$, $\gamma_k$ and the overall intercept $\alpha_0$, accounts for the residual structure in the data corresponding to all unaccounted familial and

regional factors (both genetic and environmental) that contribute to phenotypic similarities within families and within regions.

Next, to test for active gene–environment correlations, we decomposed the within-family effect into within- and between-region effects. This allowed us to investigate whether the between-family model captures all gene–environment correlations in the individual-level polygenic score or whether part of the within-family effect also contains between-region effects. We fitted the following model 5:

$$Y_{ijk} = \alpha_0 + \beta_W \left( \left( PRS_{ij} - \overline{PRS_j} \right) - \overline{\left( PRS_{ij} - \overline{PRS_j} \right)}_k \right)$$
$$+ \beta_B \overline{\left( PRS_{ij} - \overline{PRS_j} \right)}_k + \gamma_j + \gamma_k + \varepsilon_{ijk}$$

All models included sex and age as fixed covariates (not displayed in the formulas above).

To establish whether the individual-level genetic effect ($\beta_W$) differs between models 2 and 4, we need to account for the fact that we rely on near perfectly overlapping data. We do so by way of a bootstrap procedure. We resampled, with replacement, a number of observations equal to the number of observations in the original dataset from rows of the dataset. In 1,000 bootstrap samples, we re-estimated models 2 and 4, computed the difference in $\beta_W$ between models 2 and 4, and computed the standard deviation of this difference, which served as the bootstrapped standard error. Any correlation between $\beta_{W2}$ and $\beta_{W4}$ is reproduced faithfully across the bootstrap samples, meaning that the standard error of their differences accounts for this correlation. We then computed a bootstrap $P$ value assuming the distribution of $(\beta_{W2} - \beta_{W4}/\text{s.d.}(\beta_{W2} - \beta_{W4}))^2$ is $\chi^2$ with 1 degree of freedom under the null.

**Genetic association analyses.** We performed four GWASs per trait. First, we performed GWASs using two regression models. Model 1 is as follows:

$$y_i = \beta_0 + \beta_1 \times SNP + \beta_2 \times sex + \beta_3 \times age + \beta_4$$
$$\times P_1 + ... + \beta_{104} \times P_{100} + Z\mathbf{u} + e_i$$

Here $P_1$ to $P_{100}$ represent ancestry-informative PCs derived as described above, $\mathbf{u}$ is a vector of random effects and $Z$ is a random effects design matrix and a random effects model as described in ref. [17], which is used to account for the presence of closely related individuals and an extra guard against population stratification. Model 2 is as follows:

$$y_i = \beta_0 + \beta_1 \times SNP + ... + \beta_4 \times P_1 + ... + \beta_{104} \times P_{100} + \beta_{105}$$
$$\times D_1 + ... + \beta_k \times D_k + Z\mathbf{u} + e_i$$

The second equation omits sex and age for brevity and includes dummy variables $D_1$ through $D_K$ for all but one MSOA region of birth. The directed acyclic graph (DAG) in Extended Data Fig. 10a describes the suspected causal structure of the data we model, where $P$ represents the previous generation (parental influence), BP is birthplace, SNP represents the genes carried by the individual and $Y$ corresponds to the phenotypic outcome. Along similar lines, a GWAS was performed using dummy variables based on current address instead of birthplace and an analysis where dummy variables for both birthplace and current address were included. The plausible causal models in the GWASs that correct for current address are more complicated. We propose three DAGs, which we feel are abstractions of potential causal processes that relate confounding parental influences (P), genotype (SNP), current address (CA) and outcome ($Y$). We include these DAGs and their descriptions because they are useful to have in mind when evaluating our results. In practice, we expect that a mix of these processes, or more complex processes altogether, underpins the relationship between genotype and outcome.

The first causal process (Extended Data Fig. 10b) that may have a role is a process where a confounder, such as intergenerational transfer of wealth through inheritance or financial support during college (or lack thereof), is correlated to the parental genome and therefore the offspring's genome (SNP) as well as the address of the adult participant (CA), and controlling for address ensures the regression of $Y$ on SNP is no longer confounded by $P$ (similar to the process that underlies controlling for birthplace).

The second causal process (Extended Data Fig. 10c) is one where the genotype (SNP) mediated by traits, such as cognitive ability and mental health, affects people's ability to attain a higher education and/or be upwardly socially mobile, and so influences which environment people can afford to live in, which in turn influences the outcome ($Y$) of interest. Here, if we control for current address, we test for a more 'direct' effect, not mediated by environmental exposures, of genotype (SNP) on outcome ($Y$). The (probable) presence of mediation would mean that controlling for current address is likely to lead to qualitative changes in the genetic effect estimates to a different extent than when controlling for birthplace. However, even in the absence of mediation, we could expect differences herein, as current address appears to be measured more precisely.

Finally, as depicted in the causal model in Extended Data Fig. 10d, there is the risk that the genotype (SNP), through its effect on other traits (for example, cognitive ability), influences current address (CA), while the outcome ($Y$) (for example, substance use) also influences current address. In this case, conditioning on current address in a GWAS on, for example, substance use would potentially induce collider bias, which is undesirable. In practice, we do not know which of the causal processes, or which mix of causal processes, underlies the data. The effects of correcting for current address are therefore more complex to interpret.

**SNP-based heritability and genetic correlation.** Heritability and genetic correlation were estimated using LD score regression[19,20] and Genomic SEM[21] version 0.0.3. The genetic correlations represent the genetic covariation between two traits based on all polygenic effects captured by the included SNPs, and are estimated with the slope from the regression of the product of the two z-scores from the two GWASs on the LD score[20]. The genome-wide LD information used was based on European populations from the HapMap3 reference panel[19,43]. All LD score regression analyses included the ~1.3 million genome-wide HapMap SNPs used in the original LD score regression studies[19,20]. The standard error of the difference between the heritability and genetic correlations based on the different specifications could not easily be estimated because the GWASs, for which we wanted to obtain the differences were based on the exact same sample and their standard errors were therefore highly dependent. Thus, we estimated the standard error of the differences in heritability and genetic correlations with Genomic SEM, which allowed us to account for the dependence between the estimates of SNP-based heritability and genetic correlations. Genomic SEM accounts for this dependence by explicitly considering the correlation between estimates of heritability and/or genetic correlations that are modeled jointly (for details see ref. [21]). To obtain the $P$ value of the difference, we divided the estimate of the difference by its standard error and performed a $Z$ test. Significance was then determined after FDR correction of the $P$ value. For geography-corrected traits, genetic correlations were computed with either educational attainment or income, and these were also corrected for the same geographic variable.

**Reporting summary.** Further information on research design is available in the Nature Research Reporting Summary linked to this article.

## Data availability

This research was conducted using data from UK Biobank resource (application number 40310). Individual-level UK Biobank data, both phenotypic and genetic, are available to bona fide researchers on request once a research project has been submitted and approved by UK Biobank committee. Information about registration for access to the data is available at http://www.ukbiobank.ac.uk/register-apply/. The GWAS summary statistics of all 56 complex traits before and after controlling for geographic region are available through Zenodo (https://doi.org/10.5281/zenodo.6822023)[44].

## Code availability

Code for analysis is available through Zenodo (https://doi.org/10.5281/zenodo.6822023)[44].

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

## Acknowledgments

A.A. and K.J.H.V. are supported by the Foundation Volksbond Rotterdam. A.A. is also supported by ZonMw grant 849200011 from The Netherlands Organisation for Health Research and Development. M.G.N. is supported by the National Institute of Mental Health of the National Institutes of Health under award number R01MH120219, ZonMW grants 849200011 and 531003014 from The Netherlands Organisation for Health Research and Development, a VENI grant awarded by NWO (VI.Veni.191 G.030) and is a Jacobs Foundation Fellow. UK Biobank: This study has been conducted using UK Biobank resource under application number 40310. UK Biobank was established by the Wellcome Trust medical charity, Medical Research Council, Department of Health, Scottish Government and the Northwest Regional Development Agency. It has also had funding from the Welsh Assembly Government, British Heart Foundation and Diabetes UK.

## Author contributions

A.A. and M.G.N. conceived and designed the study, analyzed data and wrote the manuscript. A.A. produced the figures. C.V.D. and K.J.H.V. provided significant feedback on the methods, analyses and manuscript.

## Competing interests

The authors declare no competing interests.

## Additional information

**Extended data** is available for this paper at https://doi.org/10.1038/s41588-022-01158-0.

**Correspondence and requests for materials** should be addressed to Abdel Abdellaoui.

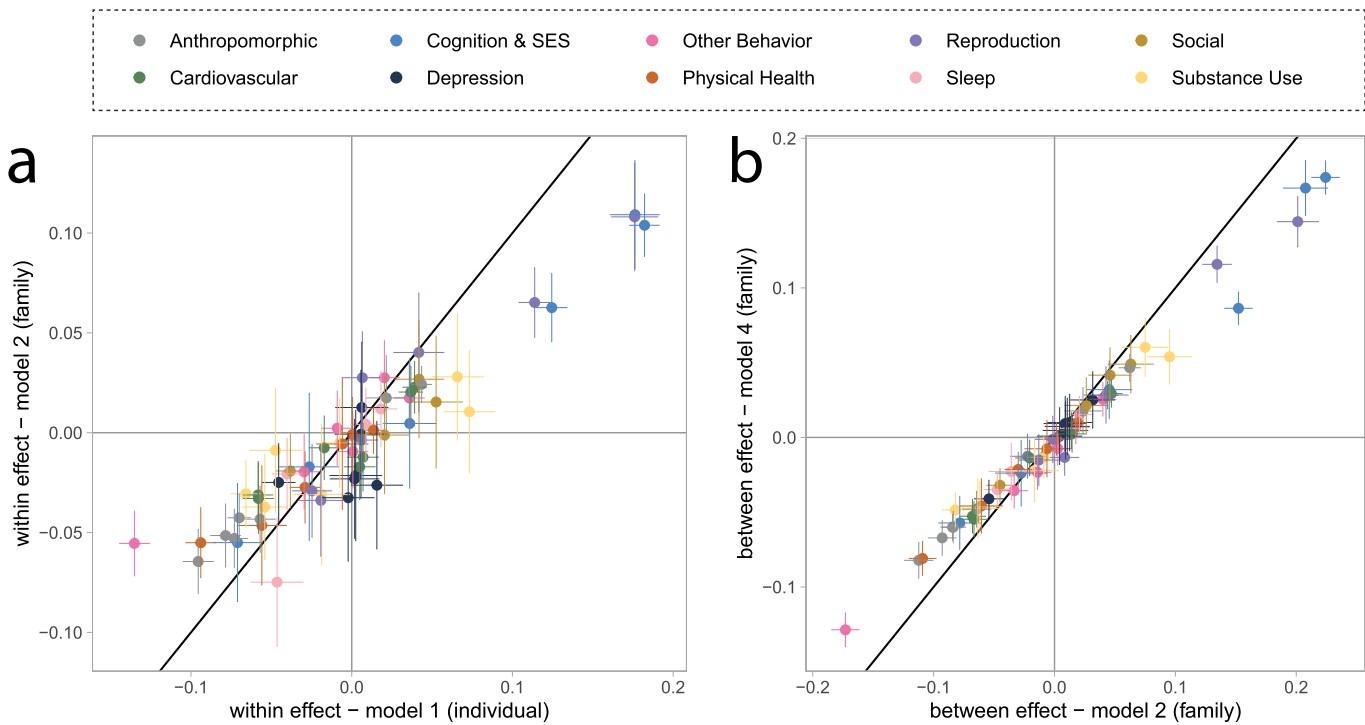

**Extended Data Fig. 1 | Results of sibling educational attainment polygenic score analyses for 56 complex traits.** Sample sizes range from 11,093 to 43,516, see Supplementary Table 1 for sample size per trait. Panel **a** shows the comparison between the individual-level effect estimate of model 1 and the within-family effect estimate from model 2. Panel **b** shows the comparison between the between-family effect estimate of model 2 and the between-family effect estimate from model 4. A total of 38 traits showed a significant decrease (FDR-corrected *P* value of the difference based on 1000 bootstraps), of which the top five are: household income (p=1.6×10$^{-103}$), educational attainment (p=1.6×10$^{-83}$), time spent watching TV (p=4×10$^{-61}$), body fat (p=7×10$^{-43}$) and BMI (p=5×10$^{-40}$).

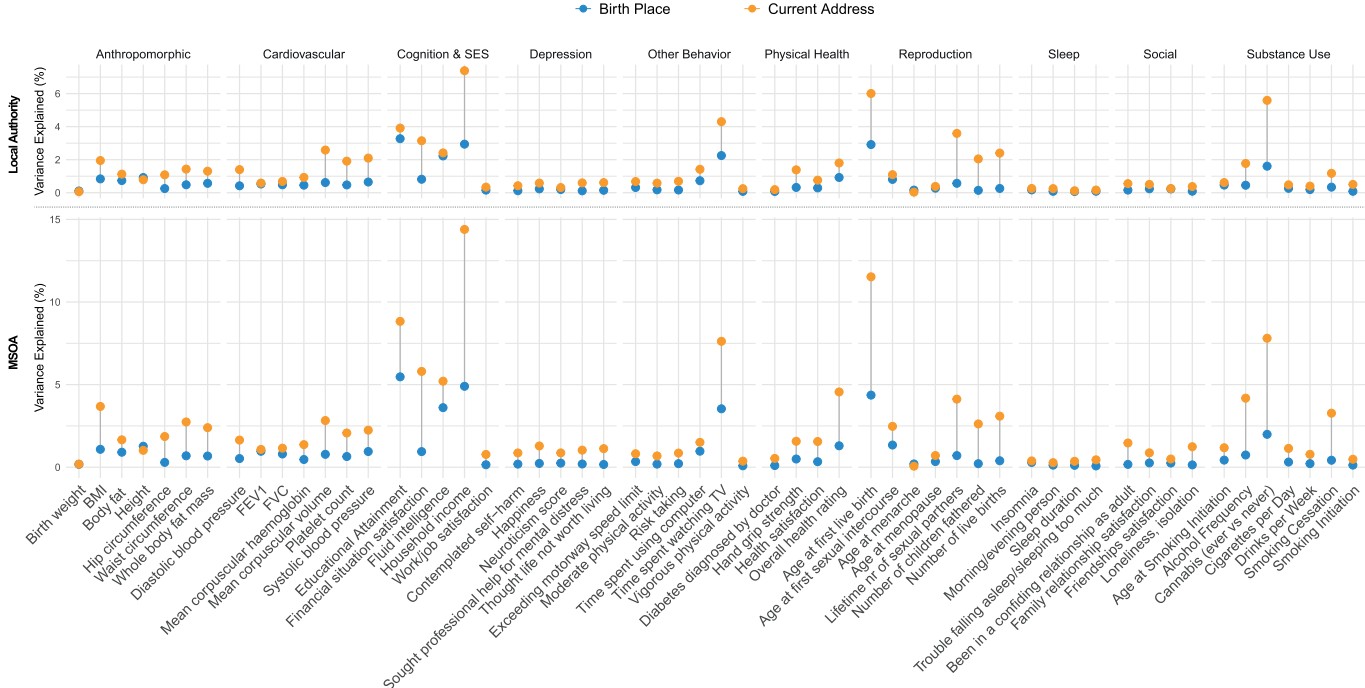

**Extended Data Fig. 2 | Variation explained by regional differences.** Linear mixed model results, with phenotype as a dependent variable and region as random effect (N = 77,111-309,387 unrelated individuals). All FDR-corrected *P* values were < 0.05.

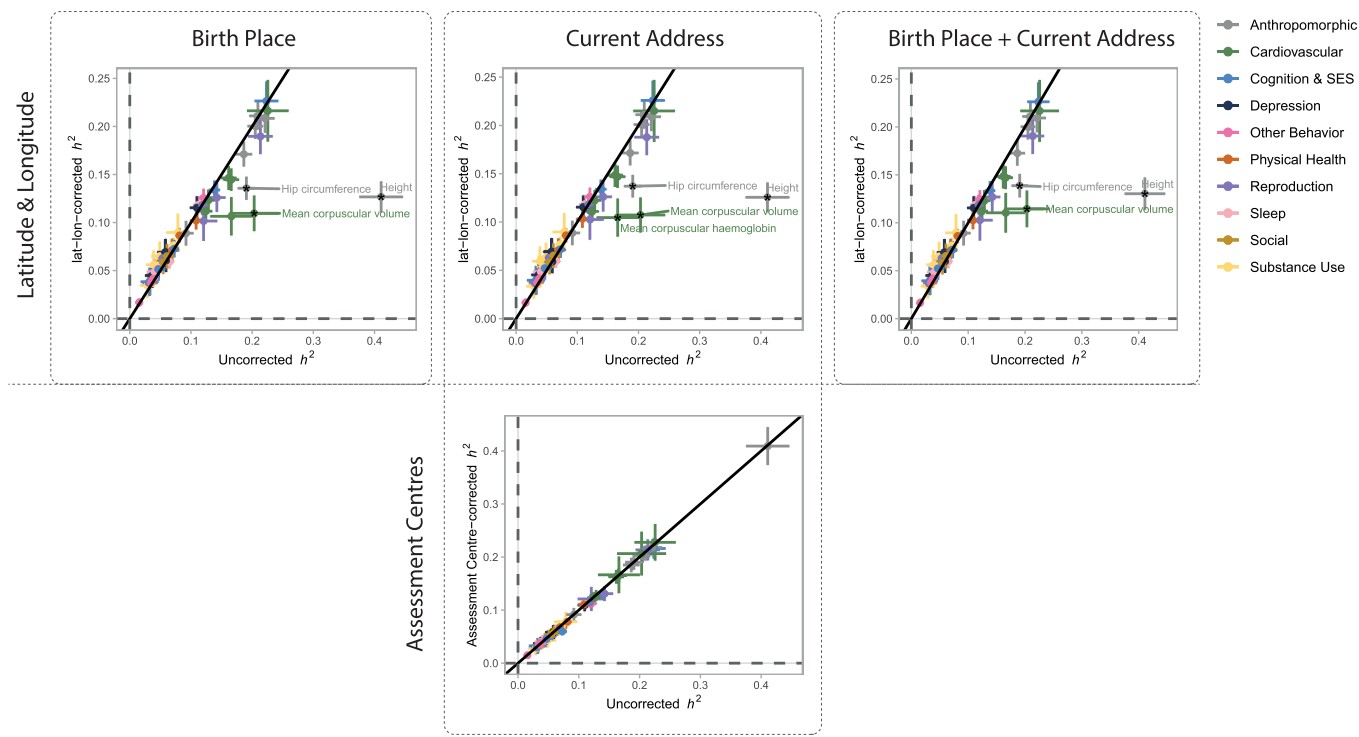

**Extended Data Fig. 3 | SNP-based heritabilities of 56 complex traits, corrected and uncorrected for geographic variables.** The panels show the estimated SNP-based heritabilities after controlling for latitude + longitude or assessment centers based on birthplace and/or current address. Error bars in indicate 95% confidence intervals. The trait names are shown for traits that showed a significant change in SNP-based heritability (FDR-corrected $P$ values < 0.05). Sample sizes of the GWASs range from 63,780 to 254,387, see Supplementary Table 1 for sample size per trait.

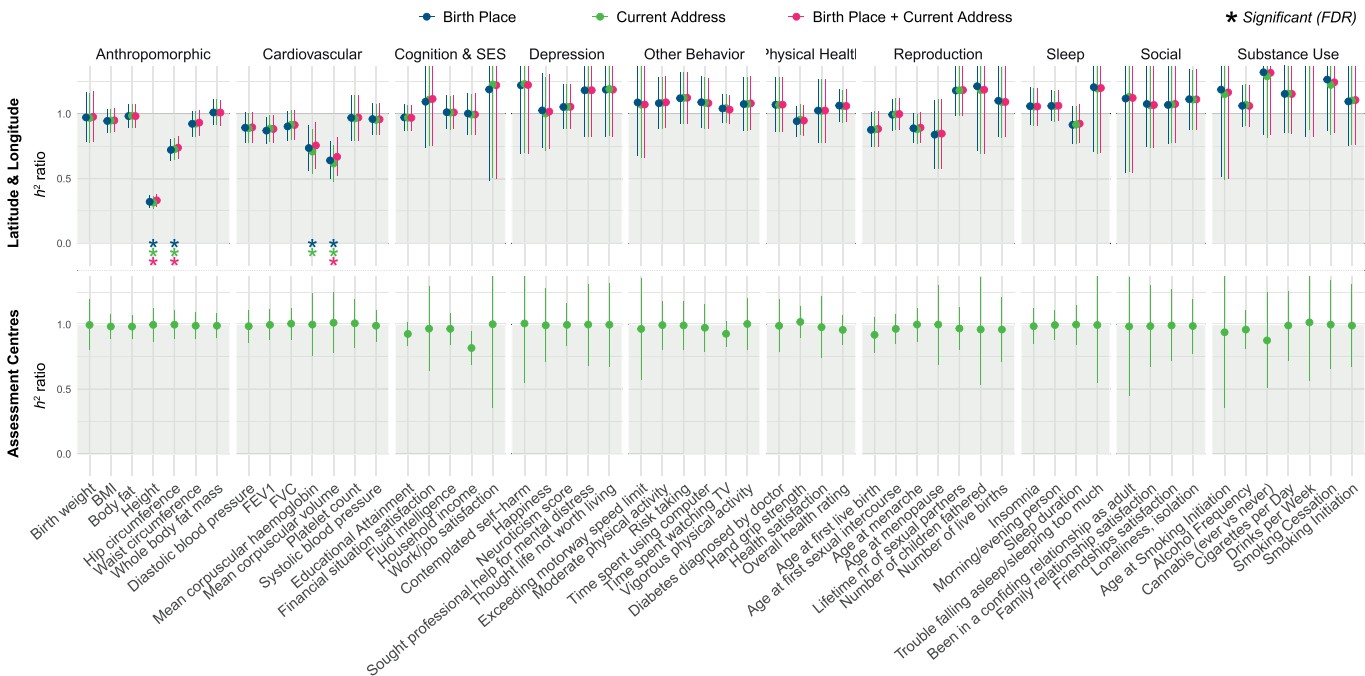

**Extended Data Fig. 4 | The ratio of decrease in SNP-based heritability after controlling for geographic variables (latitude + longitude or assessment centers) for 56 complex traits.** The ratio of the decrease is computed by dividing the corrected by the uncorrected SNP-based heritability estimate. Error bars in indicate 95% confidence intervals. The asteriks indicate FDR-corrected $P$ values < 0.05, indicating significant changes in SNP-based heritability. Sample sizes of the GWASs range from 63,780 to 254,387, see Supplementary Table 1 for sample size per trait.

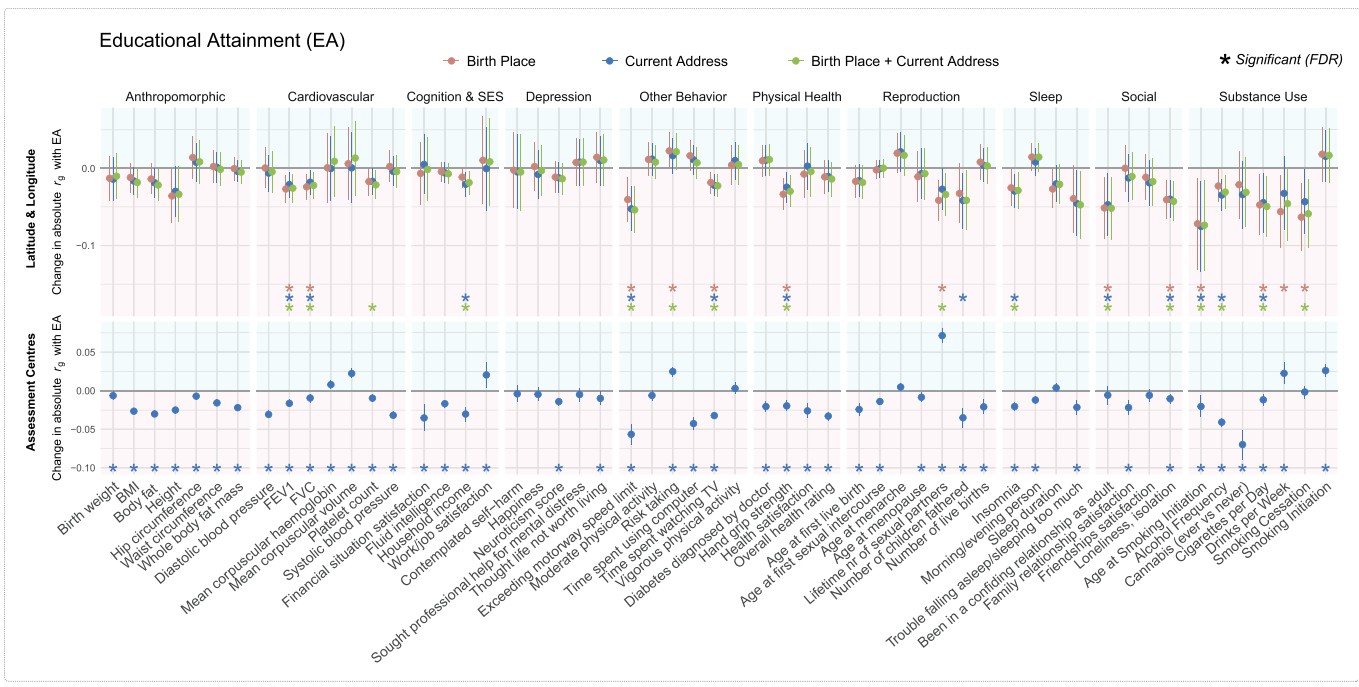

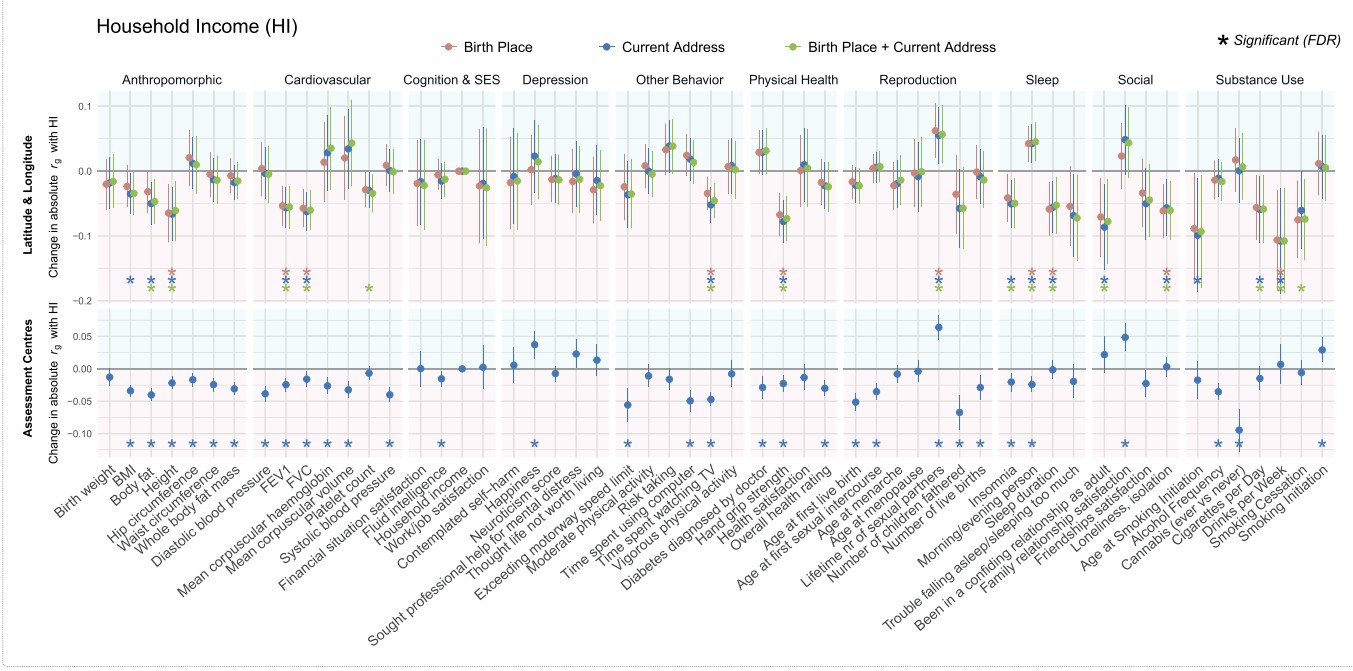

**Extended Data Fig. 5 | The change in absolute genetic correlations with educational attainment (EA, top) and household income (HI, bottom) after controlling for geographic variables (latitude + longitude or assessment centers).** The genetic correlations were computed with LDSC regression. We display the change in absolute genetic correlation to visualize the change in the strength of the genetic relationships with EA/HI (the directions of the genetic correlations vary between traits and are displayed in Extended Data Fig. 6). Error bars in indicate 95% confidence intervals. The asteriks indicate FDR-corrected *P* values < 0.05, indicating significant changes in genetic correlation. Sample sizes of the GWASs range from 63,780 to 254,387, see Supplementary Table 1 for sample size per trait.

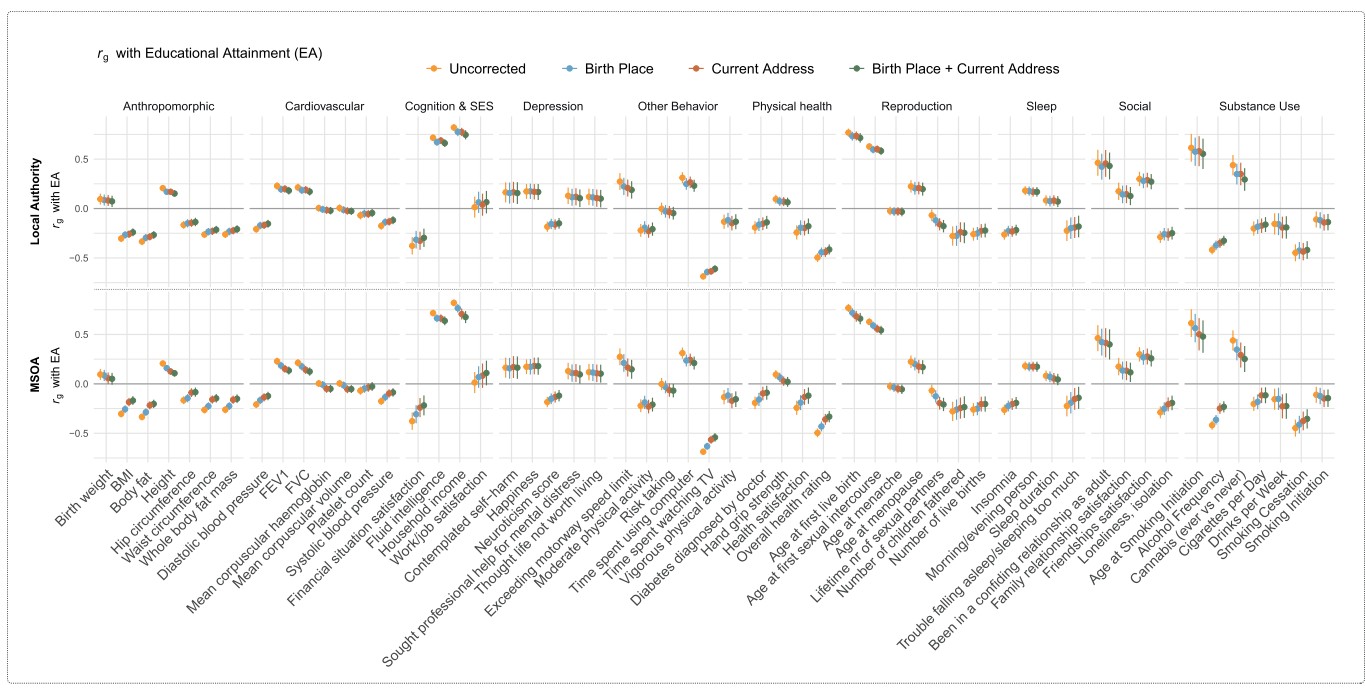

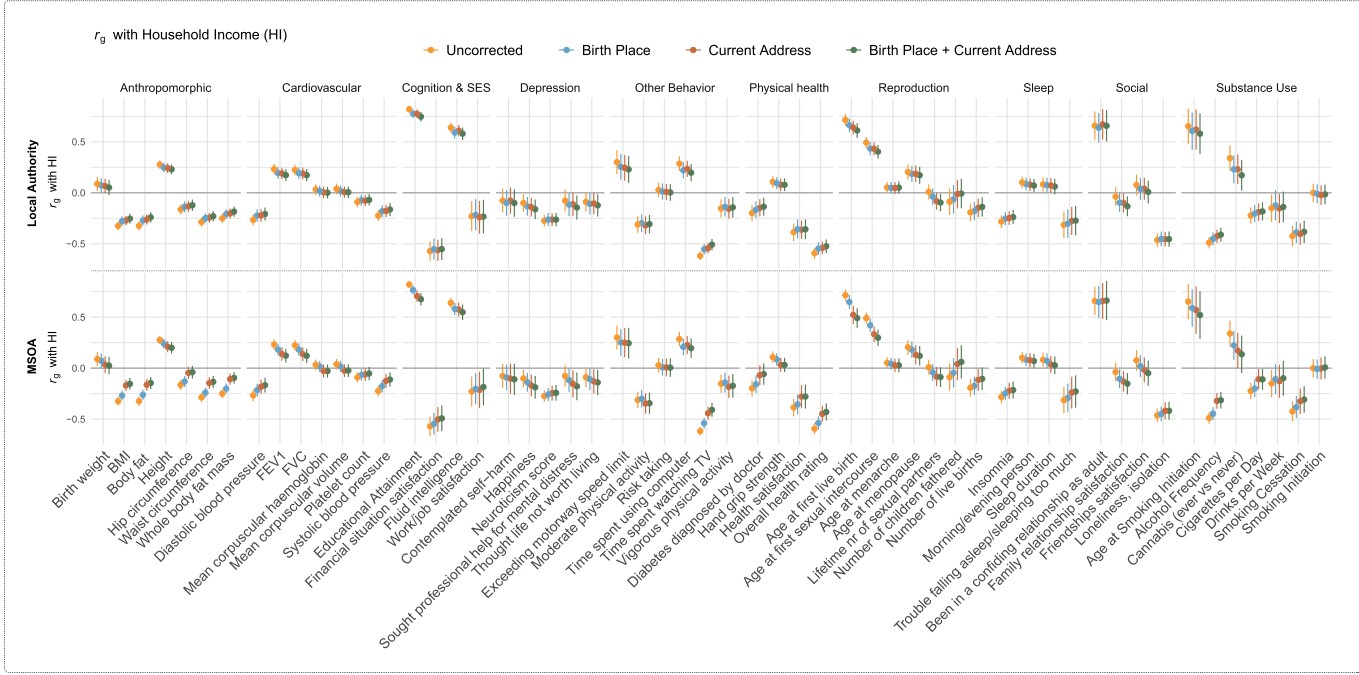

**Extended Data Fig. 6 | Genetic correlations ($r_g$) with educational attainment (EA) and household income (HI) as computed with LDSC regression, before and after controlling for Local Authority or MSOA region, based on birthplace, current address, or birthplace + current address.** Error bars in indicate 95% confidence intervals. Sample sizes of the GWASs range from 63,780 to 254,387, see Supplementary Table 1 for sample size per trait.

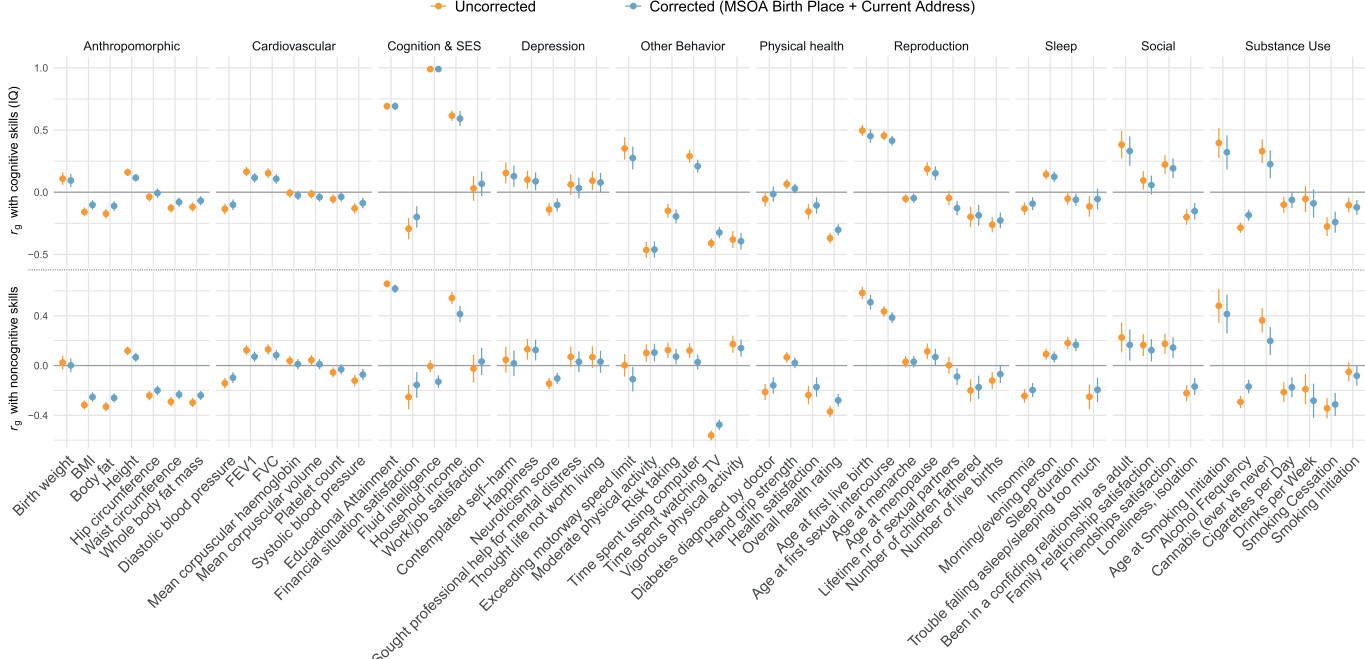

**Extended Data Fig. 7 | Genetic correlations ($r_g$), as computed with LDSC regression, with cognitive and non-cognitive skills as extracted from educational attainment GWAS by Demange et al (2019)[22], before and after controlling for MSOA region, based on birthplace + current address.** Error bars in indicate 95% confidence intervals. Sample sizes of the GWASs range from 63,780 to 254,387, see Supplementary Table 1 for sample size per trait.

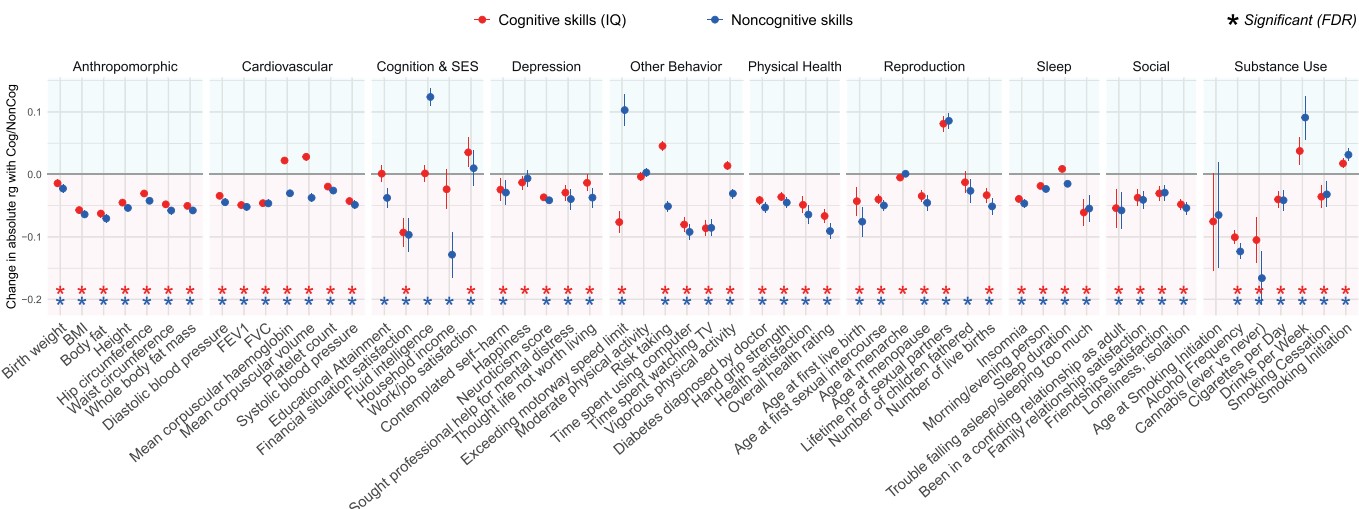

**Extended Data Fig. 8 | The change in absolute genetic correlations with cognitive and non-cognitive skills as extracted from educational attainment GWAS by Demange et al (2019)[22] after controlling for MSOA region, based on birthplace+ current address.** The genetic correlations were computed with LDSC regression. We display the change in absolute genetic correlation to visualize the change in the strength of the genetic relationships (the directions of the genetic correlations vary between traits and are displayed in Extended Data Fig. 7). Error bars in indicate 95% confidence intervals. The asteriks indicate FDR-corrected *P* values < 0.05, indicating significant changes in genetic correlation. Sample sizes of the GWASs range from 63,780 to 254,387, see Supplementary Table 1 for sample size per trait.

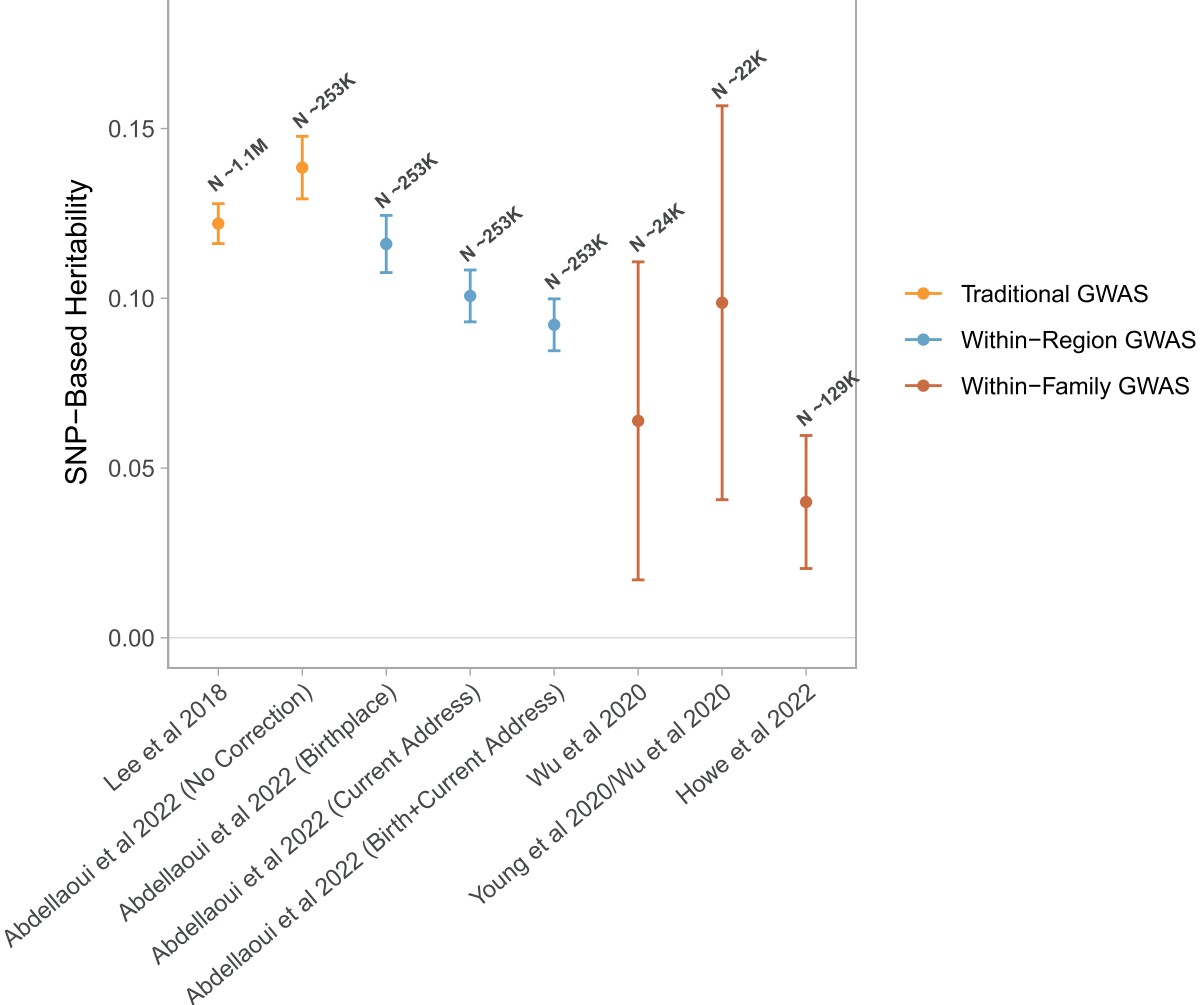

**Extended Data Fig. 9 | SNP-based heritability estimates of educational attainment (EA) under different GWAS designs.** Error bars indicate 95% confidence intervals. The sample sizes are: Lee et al [16]: N = 1,131,881; Abdellaoui et al (2022): N = 252,521; Wu et al (2020): N = 24,434; Young et al (2020)/Wu et al (2020): N = 22,207; Howe et al (2022): N = 128,777. The results from Abdellaoui et al [44] show SNP-based heritability estimates after controlling for MSOA regions.

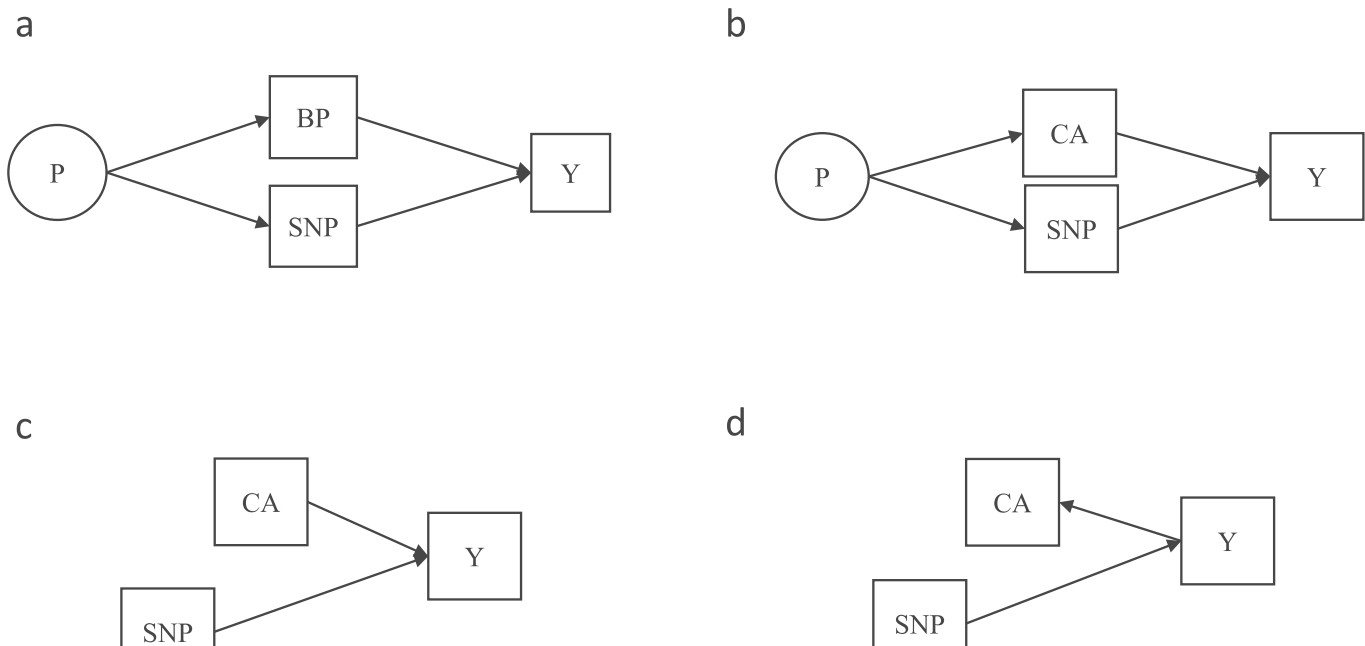

**Extended Data Fig. 10 | Directed acyclic graphs (DAGs) of the possible causal processes underlying the gene–environment correlations.** P = parental influence, BP = birthplace, CA = current address, SNP = single nucleotide polymorphism, Y = phenotypic outcome.

# Reporting Summary

Nature Research wishes to improve the reproducibility of the work that we publish. This form provides structure for consistency and transparency in reporting. For further information on Nature Research policies, see our Editorial Policies and the Editorial Policy Checklist.

## Statistics

For all statistical analyses, confirm that the following items are present in the figure legend, table legend, main text, or Methods section.

| n/a | Confirmed | |
|---|---|---|
| ☐ | ☒ | The exact sample size (*n*) for each experimental group/condition, given as a discrete number and unit of measurement |
| ☐ | ☒ | A statement on whether measurements were taken from distinct samples or whether the same sample was measured repeatedly |
| ☐ | ☒ | The statistical test(s) used AND whether they are one- or two-sided<br>*Only common tests should be described solely by name; describe more complex techniques in the Methods section.* |
| ☐ | ☒ | A description of all covariates tested |
| ☐ | ☒ | A description of any assumptions or corrections, such as tests of normality and adjustment for multiple comparisons |
| ☐ | ☒ | A full description of the statistical parameters including central tendency (e.g. means) or other basic estimates (e.g. regression coefficient) AND variation (e.g. standard deviation) or associated estimates of uncertainty (e.g. confidence intervals) |
| ☐ | ☒ | For null hypothesis testing, the test statistic (e.g. *F*, *t*, *r*) with confidence intervals, effect sizes, degrees of freedom and *P* value noted<br>*Give P values as exact values whenever suitable.* |
| ☒ | ☐ | For Bayesian analysis, information on the choice of priors and Markov chain Monte Carlo settings |
| ☒ | ☐ | For hierarchical and complex designs, identification of the appropriate level for tests and full reporting of outcomes |
| ☒ | ☐ | Estimates of effect sizes (e.g. Cohen's *d*, Pearson's *r*), indicating how they were calculated |

*Our web collection on statistics for biologists contains articles on many of the points above.*

## Software and code

Policy information about availability of computer code

| Data collection | No software was used in the data collection. |
|---|---|
| Data analysis | The open source software used for this study was: GCTA 1.93, PLINK 2.0, flashPCA v2, LDSC Regression 1.0, Genomic SEM 0.0.3, and R 1.4.1717. Our code used for the analyses is available through Zenodo (https://doi.org/10.5281/zenodo.6822023). |

For manuscripts utilizing custom algorithms or software that are central to the research but not yet described in published literature, software must be made available to editors and reviewers. We strongly encourage code deposition in a community repository (e.g. GitHub). See the Nature Research guidelines for submitting code & software for further information.

## Data

Policy information about availability of data

All manuscripts must include a data availability statement. This statement should provide the following information, where applicable:
- Accession codes, unique identifiers, or web links for publicly available datasets
- A list of figures that have associated raw data
- A description of any restrictions on data availability

This research was conducted using data from the UK Biobank resource (application number 40310). Individual-level UK Biobank data, both phenotypic and genetic, is available to bona fide researchers on request once a research project has been submitted and approved by the UK Biobank committee. Information about registration for access to the data is available at http://www.ukbiobank.ac.uk/register-apply/. The GWAS summary statistics of all 56 complex traits before and after controlling for geographic region are available through Zenodo (https://doi.org/10.5281/zenodo.6822023).

# Field-specific reporting

Please select the one below that is the best fit for your research. If you are not sure, read the appropriate sections before making your selection.

☒ Life sciences ☐ Behavioural & social sciences ☐ Ecological, evolutionary & environmental sciences

For a reference copy of the document with all sections, see nature.com/documents/nr-reporting-summary-flat.pdf

# Life sciences study design

All studies must disclose on these points even when the disclosure is negative.

| | |
|---|---|
| Sample size | The sample size was determined by UK Biobank; see for more details Bycroft, C. et al (2018). The UK Biobank resource with deep phenotyping and genomic data. Nature 562, 203. |
| Data exclusions | In order to avoid bias due to populations stratification, we excluded participants with non-European ancestry as determined by PCA on genome-wide SNPs (details are described in the Online Methods). |
| Replication | The current study was successful because of the exceptionally large sample size of UK Biobank, allowing for sufficient participants per geographic region in the UK. There are no genotype datasets of similar size and geographic spread as this one, making replication not (yet) feasible. |
| Randomization | The participants were chosen across 22 assessment centers throughout Great Britain in order to cover a variety of different settings providing socioeconomic and ethnic heterogeneity and urban–rural mix. (see for more details Bycroft, C. et al (2018). The UK Biobank resource with deep phenotyping and genomic data. Nature 562, 203). We investigated the impact of this non-random ascertainmet by including assessment centre as a covariate, and concluded that its impact was not substantial (see our manuscript for more details). |
| Blinding | Blinding was not relevant as this was not an experimental design, and therefore did not have experimental groups; we included everyone in our analyses from this population-based sample with genotype and phenotype data available. |

# Reporting for specific materials, systems and methods

We require information from authors about some types of materials, experimental systems and methods used in many studies. Here, indicate whether each material, system or method listed is relevant to your study. If you are not sure if a list item applies to your research, read the appropriate section before selecting a response.

## Materials & experimental systems

| n/a | Involved in the study |
|---|---|
| ☒ | ☐ Antibodies |
| ☒ | ☐ Eukaryotic cell lines |
| ☒ | ☐ Palaeontology and archaeology |
| ☒ | ☐ Animals and other organisms |
| ☐ | ☒ Human research participants |
| ☒ | ☐ Clinical data |
| ☒ | ☐ Dual use research of concern |

## Methods

| n/a | Involved in the study |
|---|---|
| ☒ | ☐ ChIP-seq |
| ☒ | ☐ Flow cytometry |
| ☒ | ☐ MRI-based neuroimaging |

# Human research participants

Policy information about studies involving human research participants

| | |
|---|---|
| Population characteristics | A total of 502,536 participants (273,402 females and 229,134 males) aged between 37 and 73 years old were recruited in the UK between 2006 and 2010. The participants were recruited across 22 assessment centers throughout Great Britain in order to cover a variety of different settings providing socioeconomic and ethnic heterogeneity and urban–rural mix. This is a population-based sample, wherein participants were not chosen based on current or past diagnoses. See for more details Bycroft, C. et al (2018). The UK Biobank resource with deep phenotyping and genomic data. Nature 562, 203. |
| Recruitment | The UK Biobank ascertainment strategy was designed to capture sufficient variation in socioeconomic, urban–rural, and ethnic background. The participation rate however was 5.45% and was biased towards older, more healthy, and female residents. The UK Biobank sample does reflect nationally representative data sources to a significant degree. |
| Ethics oversight | The participants of this study come from UK Biobank (UKB), which has received ethical approval from the National Health Service North West Centre for Research Ethics Committee (reference: 11/NW/0382). |

Note that full information on the approval of the study protocol must also be provided in the manuscript.

