## [Peer Review File · Nature Genetics]

Peer Review Information

Manuscript Title: Gene-environment correlations across geographic regions affect genome-wide association studies

Corresponding author name(s): Dr Abdel Abdellaoui

Reviewer Comments & Decisions:

Decision Letter, initial version:

24th May 2021

Dear Dr Abdellaoui,

Your Article, "Geographic Confounding in Genome-Wide Association Studies" has now been seen by 2 referees. You will see from their comments below that while they find your work of interest, some important points are raised. We are interested in the possibility of publishing your study in Nature Genetics, but would like to consider your response to these concerns in the form of a revised manuscript before we make a final decision on publication.

As you will see from these comments, both referees have identified aspects of the analyses and the discussion that should be improved. In addition, reviewer #2 has concerns regarding the interpretation of "geographic confounding" and the use of the term "gene-environment correlation". Please address all referees points as thoroughly as you can.

We therefore invite you to revise your manuscript taking into account all reviewer and editor comments. Please highlight all changes in the manuscript text file. At this stage we will need you to upload a copy of the manuscript in MS Word .docx or similar editable format.

*2) If you have not done so already please begin to revise your manuscript so that it conforms to our Article format instructions, available

[here](http://www.nature.com/ng/authors/article_types/index.html).

*3) Include a revised version of any required Reporting Summary:

[REDACTED]

We hope to receive your revised manuscript within three to six months. If you cannot send it within this time, please let us know.

Sincerely,

Wei Li, PhD
Senior Editor
Nature Genetics

2One New York Plaza, 47th Fl.
New York, NY 10004, USA
www.nature.com/ng

Reviewers' Comments:

Reviewer #1:

Remarks to the Author:

This is a well-written and interesting paper from a strong team of investigators. The general problem of confounding in observational data is an extremely important issue. Due to its large size and scope, and wide use within the biomedical research community, the UK Biobank is a critical dataset in which to study the effects of confounding.

I take no issue with the analytic choices made in the manuscript, which all appear generally reasonable even if another team may make somewhat different choices or conduct additional sensitivity checks and so on. At the same time, as I suppose is almost always true, there were issues about which I was confused or had what I hope are constructive suggestions.

1. I very much appreciated and found refreshing that the authors explicitly stated the causal scenario(s) they were assuming throughout the manuscript. The million dollar question (for me) is whether the geographical covariates represent confounders or something else. The present data and analyses allow one to evaluate how parameters change in this dataset when more covariates are added, but do not seem to provide a great deal of information about whether those covariates truly are confounders, colliders, or whether a real genetic effect is being eliminated through a misunderstood covariate (e.g., the SNP causes the location which then causes the phenotype; the authors' scenario B). This is stated in the methods quite directly (line 459) but would seem to affect many of the conclusions described in the discussion.

2. What do these geolocation variables measure about these individuals? (SES indicators such as education, income, and Townsend are available to do some psychometrics of the geographic info I assume?) What is measured would be highly relevant to interpretation of the results, and under what circumstance or for what scientific questions one should include these geographic covariates.

3. Related to the above, while I recognize that no single paper can do it all, I remain curious to know how the authors would include UKB ascertainment in their causal diagrams and whether/how that issue further muddies the causal waters that are central to the paper's conclusions that the geographic measures are confounders. Perhaps this is too far afield.

4. Would the authors please provide a more explicit argument as to why geographic confounding as studied here is distinct from population stratification? At the very least, these two things would seem to be linked, perhaps inextricably at least in these data.

35. I'd like the authors to further convince the reader that birthplace is truly qualitatively distinct from current address. Aren't the parents' current address at the time of conception potentially influenced by the same genetic variants that are passed on to the child, such that an inherited SNP or genetic predisposition could cause the birthplace in some sense? I'll be honest I find it difficult to think through, and I imagine readers will as well.

6. This last comment is simply food for thought, I am not suggesting the authors necessarily take this on. The authors are attempting to understand gene-environment correlation between geographical location (as a proxy for whatever) and a wide range of phenotypes. Can you use migration information (e.g., the change between birth and current location) to help disentangle the relative roles of genes and environment affecting the development of a phenotype? Surprisingly, on a brief literature search I was unable to find such a study in the UK Biobank.

I am in the habit of signing my reviews,

Scott Vrieze

Reviewer #2:

Remarks to the Author:

Abdellaoui et al. investigate and interpret the impact of correcting for geographic region (e.g. current address), a genetically heritable trait, in GWAS of educational attainment and related traits.

Major comments:

1. I disagree with the interpretation of "geographic confounding" that is provided throughout this paper. Although this is more of a disagreement with the interpretation than a disagreement with the scientific analyses, it still represents a fundamental disagreement.

In detail, the observation that geographic region is a genetically heritable trait (after correcting for genetic ancestry) was made by Abdellaoui 2019 NHB, which is appropriately cited. The interpretation of Abdellaoui 2019 NHB is that migration is partly driven by SES, which is genetically heritable. Notably, that paper did not ever use the word "confounding". Instead, it simply stated that geographic region is a genetically heritable trait. Thus, it seems to me that the current paper is partitioning the genetically heritable component of educational attainment into the part that can be explained by geographic region and the part that cannot be explained by geographic region. This is analogous to partitioning the genetically heritable component of educational attainment into the part that can be explained by cognitive ability and the part that cannot be explained by cognitive ability (Demange 2021 NG; same last author as this paper); that paper also did not ever use the word "confounding". The same analogy applies to the example on p.3 of the current paper involving a scenario of more expensive insulin leading to a genetic correlation between educational attainment and untreated diabetes. In that case, one could partition the genetically heritable component of untreated diabetes

4into the part that can be explained by educational attainment and the part that cannot be explained by educational attainment.

I also disagree with the term "gene-environment correlation", because geographic region is not a solely environmental variable, but rather a genetically heritable trait, as shown by Abdellaoui 2019 NHB. Indeed, the demonstration in the current paper that current address is more important than birthplace provides compelling proof that geographic region is genuinely influenced by the target individual's genetics, as opposed to genetic ancestry or parental genotype effects.

Thus, it is important to either (a) agree with me and drop the use of the term "confounding" throughout the paper, or (b) disagree with me but include a balanced presentation of my point of view in the Introduction and Discussion sections, while providing a compelling explanation of why my point of view is not preferred.

Would NG readers be interested to see another partitioning of the genetically heritable component of educational attainment, beyond that previously provided by Demange 2021 NG? I suggest that the answer could potentially be yes, as this partitioning of the genetically heritable component of educational attainment is completely different and unrelated to that provided by Demange 2021 NG, and geographic region is an interesting genetically heritable trait.

2. Demange 2021 NG computed and analyzed summary statistics for the non-cognitive part of educational attainment as a linear combination of (i) summary statistics for educational attainment and (ii) cognitive ability. Would it likewise be possible in the current paper to compute and analyze summary statistics for the part of educational attainment that can be explained by geographic region as a linear combination of summary statistics for (i) educational attainment and (ii) the part of educational attainment that cannot be explained by geographic region (both of which are analyzed in the current paper)? It seems to me that the answer should be yes, or perhaps this partitioning could even be done at the level of individual-level data. This is an important question that must be addressed in the paper. Specifically, either the part of educational attainment that can be explained by geographic region should be included in the SNP-based heritability and genetic correlation analyses, or the paper should provide a compelling explanation as to why this is not possible.

3. What would happen if you corrected only for latitude and longitude of current address and/or birthplace, instead of dummy variables representing 859-1,959 distinct regions. Would you get similar results? I suggest that this is an important analysis that should be performed. If it would turn out that correcting for latitude and longitude would be sufficient to produce similar results (although I doubt this), that would be far simpler and thus preferred in future research projects.

4. What would happen if you corrected for dummy variables representing a much smaller number of distinct regions (e.g. 10 regions, instead of 859-1,959 distinct regions). Would you get similar results? I suggest that this is an important analysis that should be performed. If it would turn out that correcting for dummy variables representing a much smaller number of distinct regions would be sufficient to produce similar results (although I doubt this), that would be far simpler and thus preferred in future research projects.

5. In the genetic correlation analysis, I suggest that it would of interest to also include the following traits: (a) educational attainment in non-UK Biobank European samples (see e.g. Demange 2021 NG), (b) the cognitive component of educational attainment and the non-cognitive component of educational attainment (see Demange 2021 NG).

6. The paper states that correcting for current address has a larger a greater impact than correcting for birthplace. This seems to be a reasonable statement based on the information that is provided (#traits with significant change in SNP-based heritability or genetic correlation, when correcting for birthplace or current address or birthplace + current address). However, as this is an interesting result, a more precise quantification would be preferred. Specifically, in both SNP-based heritability and genetic correlation analyses, for how many traits is the impact of correcting for current address significantly larger than the impact of correcting for birthplace, at an appropriate and explicitly specified significance level?

Minor comments:

-- p.2: "reflective of a causal effect of that allele": this statement does not account for the widely acknowledged effects of LD-tagging, and should be adjusted accordingly.

-- p.2: population stratification is described as an entirely environmental phenomenon. However, it can also be partially or entirely genetic. (For example, one could envision a hypothetical scenario in which all differences in height between human populations are genetically driven. This could still lead to false-positive associations at individual loci that do not causally impact height.) Please modify the text to note this possibility.

-- p.2: population stratification: consider citing/discussing Haworth 2019 NC, which pertains specifically to the UK Biobank data set analyzed in this study.

-- p.3: "geographic confounding has been shown to exist [Abdellaoui 2019 NHB]". This statement is incorrect, as the paper of Abdellaoui 2019 NHB does not include the word "confounding". See Major comment 1. Please modify the text accordingly.

-- p.3: I suggest that it would be good to clarify here why only a subset of 271,457 of the 456,064 European-ancestry samples were included. Was it primarily because latitude and longitude was available for only a subset of European-ancestry samples? Or primarily because discarding MSOA regions with <100 samples caused many samples to be excluded?

-- p.4: "The LMM GWAS controls for cryptic relatedness and population stratification by including a genetic relatedness matrix (GRM) in the model [Jiang 2019 NG]". This statement is incorrect. In the method of Jiang 2019 NG, PCs are used to correct for population stratification, and a sparse GRM is used to correct for cryptic relatedness. Because sparsity is imposed on the GRM, the GRM does not correct for population stratification. (The word "additional" in the following sentence is also incorrect.) Please modify the text accordingly.

-- p.4: "see Supplementary Table 1 for full list of traits and sample sizes": the NG submission included two supplementary table files, one with SNP-based heritability estimates and one with genetic correlation estimates. Neither of these files included sample size information, which should be included.

-- p.5: "significant decrease": please state in this paragraph the criteria for significance of the decrease that is employed in this paragraph.

-- p.5: what is the interpretation for these decreases? It may be preferred to provide some interpretation of the results in the Results section, instead of exclusively in the Discussion section.

6

- p.6: "Genes associated with socio-economic success can have an influence on the neighborhood that people can afford to live in": although I find this to be a compelling statement (and a further affirmation of my Major comment 1), this statement is not specific to genetic correlations, and thus I suggest that it should be moved to earlier than the genetic correlations section, which should focus on genetic correlations.
- p.6: "significant change (FDR corrected)": please state in this paragraph the precise significance threshold.
- p.7: what is the interpretation of the significantly weaker genetic correlations? It may be preferred to provide some interpretation of the results in the Results section, instead of exclusively in the Discussion section.
- p.7: I disagree with the statement that within-family analyses would remove the part (of the genetically heritable component of educational attainment) that is due to geographic region. Note that current address, which is shown in this paper to be more important than birthplace, differs among adult siblings. Either this statement should be removed, or this point of disagreement with this statement should be discussed.
- p.15: For the genetic correlations with educational attainment that were adjusted for geographic region, were these estimated by analyzing summary statistics for (i) educational attainment adjusted for geographic region and (ii) the second trait, or by analyzing summary statistics for (i) educational attainment adjusted for geographic region and (ii) the secondary trait adjusted for geographic region? I am guessing that this distinction would make little difference, but it is still appropriate to be clear about exactly which analysis was performed.
- Figure 2: consider restricting to a subset of 10 or fewer traits (it is not mandatory for main Figures to include all results of all analyses, which can be provided in Supp Tables), and labeling the most interesting points in panels A-C by color-coded trait name, rather than trait category.
- Figure 3: consider restricting to a subset of 10 or fewer traits (it is not mandatory for main Figures to include all results of all analyses, which can be provided in Supp Tables).
- Figure 4: while I agree that within-family GWAS are less powerful at current sample sizes, I find this to be an obvious point that is more deserving of a Supp Figure than a main Figure for visualization.
- Data availability: the authors should publicly release summary statistics for educational attainment and household income, both uncorrected and corrected for current address and/or birthplace. Public release of summary statistics is definitely permitted by UK Biobank.
- Data availability: it would be even better if individual-level data for current address and/or birthplace (859-1,959 distinct regions as inferred in this paper) could be released through UK Biobank, although I am unsure about the mechanism for doing so.

Author Rebuttal to Initial comments

Reviewers' Comments:

Reviewer #1:

Remarks to the Author:

This is a well-written and interesting paper from a strong team of investigators. The general

7problem of confounding in observational data is an extremely important issue. Due to its large size and scope, and wide use within the biomedical research community, the UK Biobank is a critical dataset in which to study the effects of confounding.

I take no issue with the analytic choices made in the manuscript, which all appear generally reasonable even if another team may make somewhat different choices or conduct additional sensitivity checks and so on. At the same time, as I suppose is almost always true, there were issues about which I was confused or had what I hope are constructive suggestions.

We thank the reviewer for the positive evaluation of our manuscript and for his constructive feedback. We agree that it is important to disentangle the sources of variation in a widely used dataset as UK Biobank, and we also hope to encourage researchers to look for similar effects in other datasets.

We are happy with the constructive suggestions below and have, to the best of our abilities, incorporated all suggestions, which we believe have substantially improved our manuscript.

1. I very much appreciated and found refreshing that the authors explicitly stated the causal scenario(s) they were assuming throughout the manuscript. The million dollar question (for me) is whether the geographical covariates represent confounders or something else. The present data and analyses allow one to evaluate how parameters change in this dataset when more covariates are added, but do not seem to provide a great deal of information about whether those covariates truly are confounders, colliders, or whether a real genetic effect is being eliminated through a misunderstood covariate (e.g., the SNP causes the location which then causes the phenotype; the authors' scenario B). This is stated in the methods quite directly (line 459) but would seem to affect many of the conclusions described in the discussion.

We agree with the Reviewer that, in the previous version of the manuscript, there is an underlying assumption of the scenario of passive and active gene-environment correlations underlying our observations. This was mainly driven by observations made in previous work where we showed that migration induces geographic clustering of educational attainment polygenic scores in such a way that they line up with regional differences in SES, which we then show to correlate with a range of environmental factors in Abdellaoui et al (2019; *Genetic correlates of social stratification in Great Britain, Nature Human Behaviour*).

We have now substantially extended the analyses in our article by testing that assumption again with a different approach in a sib-design. We used the approach that was implemented by Selzam et al

(2019; *Comparing Within- and Between-Family Polygenic Score Prediction, AJHG*) to show the presence of gene-environment correlations on a family level. Selzam et al performed their analyses in children and young adolescents, aged 12 to 21, where most of the gene-environment correlations are expected to originate from the family environment in which they were born and raised, i.e., passive gene-environment correlations. We conduct the analyses in 22,675 adult sibling pairs from UK Biobank, aged 40 to 70. We extended the analyses by including not only within-family fixed effects, but also within-region fixed effects. Our results imply that regional-level gene-environment correlations can affect polygenic signals beyond the inflation that occurs on a family-level, in part due to active gene-environment correlations. We describe the results in the new *Results* paragraphs “Part 1: Polygenic Scores in Siblings”, Table 1 and Figures 1 & 2.

2. What do these geolocation variables measure about these individuals? (SES indicators such as education, income, and Townsend are available to do some psychometrics of the geographic info I assume?) What is measured would be highly relevant to interpretation of the results, and under what circumstance or for what scientific questions one should include these geographic covariates.

We have addressed this issue to some extent in our previous article on the geographic distribution of genetic variation (Abdellaoui et al, 2019), and have now expanded on this in the revision of the current article with new analyses and a new Figure. These geographic variables seem to most strongly capture the increasing geographic clustering of SES in Great Britain, more so when based on current address.

In Abdellaoui et al (2019; *Genetic correlates of social stratification in Great Britain, Nature Human Behaviour*) we used a regional GWAS (RGWAS) approach to investigate how well they capture regional differences associated with socio-economic, health, and cultural outcomes. In this approach, all UK Biobank participants from the same region got the same phenotype assigned, and the phenotypes were obtained from public census databases (i.e., not from UK Biobank itself). These RGWASs were performed for outcomes on both local authority and MSOA level. Nearly all RGWAS outcomes showed extremely high genetic correlations with educational attainment. Even when analysing regional differences in height or BMI, the genetic correlation was $\sim .9$ with educational attainment, and $< .3$ with height or BMI (see Extended Data Figure 9 of Abdellaoui et al 2019). The polygenic score for Educational attainment also showed strongest geographic clustering, which was strongly in line with regional SES, as captured by measures like Townsend and regional educational levels. In addition, we also computed intraclass correlations in that article on polygenic scores and ancestry-informative PCs, for both local authorities and MSOA regions. The largest amount of variation explained by these regions was for the polygenic score for educational

attainment (see Extended Data Figure 2 of Abdellaoui et al, 2019). Furthermore, current address explained substantially more variation than birth place, in line with growing clustering due to SES-driven migration. Another strong piece of evidence for the migration-hypothesis is that the region explained a *decreasing* amount of variation for ancestry-informative principal components (PCs) when looking at current address versus birth place (see Extended Data Figure 3 of Abdellaoui et al, 2019), while the clustering for polygenic scores *increased*. We strongly suspect that this can be explained by the fact that geographic distributions of older ancestry differences would get disrupted by the more recent migrations that increase geographic clustering for SES-related genetic variation.

For the current article, we have now added intraclass correlation analyses that estimate the explained variance by both local authority and MSOA regions for all 56 complex traits that were analysed in this study. The results are displayed in Supplementary Figure 1, and explained in the results section (first section in the paragraph “Part 2: GWASs Controlled for Geography”) as follows:

“Both the local authority regions and MSOA regions explained significant amounts of variation for all 56 complex traits, most for Household Income (~5% based on birthplace, ~14% based on current address; see Supplementary Figure 1). When regions were based on current address, they consistently explained more variation than when they were based on birth place, indicating that richer and more healthy people are more likely to have moved to regions inhabited by richer and more healthy people.”

As expected, most variation is explained for traits closely related to SES, and more so for current address than for birth place. What wasn't clear from the analyses in Abdellaoui et al 2019, but is clearly visible in Supplementary Figure 1, is that income shows much stronger regional differences than educational attainment (almost 15% of the variance of income is explained by MSOA based on current address, as opposed to ~9% for educational attainment). Income was not included in Abdellaoui et al 2019, because that article analysed polygenic scores in UK Biobank, and since there was no large GWASs available for income that excluded UK Biobank, we were not able to build a polygenic score for income.

The reviewer rightly asks under what circumstances these geographic covariates should be included. Our results imply that their effects will be greatest when analysing variables directly related to SES, such as educational attainment and income. However, they will affect a very broad range of other traits as well, since the reach of the effects of SES are so broad, affecting both physical and mental health outcomes and normal variation, such as reproduction or how people spend their leisure time (e.g., watching TV). This is supported by our results, given the significant reduction of genetic correlations with educational attainment and income for the majority of traits investigated.

3. Related to the above, while I recognize that no single paper can do it all, I remain curious to know how the authors would include UKB ascertainment in their causal diagrams and whether/how that issue further muddies the causal waters that are central to the paper's conclusions that the geographic measures are confounders. Perhaps this is too far afield.

We agree with the reviewer that the influence of ascertainment bias is worth considering as well as difficult to fully disentangle. A substantial part of spatial ascertainment bias will be related to the geographic location of the assessment centres. In order to investigate the impact of this, we have now added analyses that control for the assessment centres. The results are discussed in the text (Results paragraphs *SNP-Based Heritability* and *Genetic Correlations*) and displayed in Supplementary Figures 2-4. Controlling for assessment centres affects the heritability estimates and genetic correlations with SES outcomes substantially less than the local authority and MSOA regions, which suggests that this geographic bias is at least not causing the presence of SES-related polygenic signals in the uncorrected GWASs of the 56 traits analysed.

Another source of ascertainment bias is, like most biobanks, an overrepresentation of higher SES participants. In Abdellaoui et al 2019, we have a section in the Supplementary Material titled “Ascertainment Bias”, in which we address this issue by comparing census data with UK Biobank data. We conclude here that the geographic clustering of SES is likely to be stronger in the general population than captured by UK Biobank, where substantial portions of the lower SES population are missed. We now mention this in the discussion of the current article as follows: “*It was previously shown that the geographic clustering of SES is possibly stronger in the general population than in UK Biobank,¹⁴ suggesting that the effects observed in the current study may be an underestimation of the true effects.*”.

4. Would the authors please provide a more explicit argument as to why geographic confounding as studied here is distinct from population stratification? At the very least, these two things would seem to be linked, perhaps inextricably at least in these data.

We agree with the reviewer that it is not always easy to distinguish variation due to population stratification from variation due to complex trait variation, also because these sometimes overlap. In general, however, population stratification is related to older systematic ancestry differences, which are generally captured well by PCs and are largely due to genetic drift and perhaps partly due to diversifying natural selection pressures that target complex trait variation, hence the occasional overlap. We have previously gone to great lengths to show that, geographically, the regional differences that remain after controlling for these ancestry differences are related to actual complex

trait variation (Abdellaoui et al, 2019; see the paragraph in the Supplementary Material of this paper titled “*Population Stratification*”), which mostly seem to be genes associated with educational attainment, which is possibly just a good proxy for the collection of traits that make it easier or more difficult to retain or change SES in contemporary (British) society. These “functional” genes then cluster geographically with a range of environmental exposures that vary with SES, and this is what we control for in the current article.

We have now clarified the distinction more clearly in the revised introduction. The second section of the introduction now gives a better explanation of population stratification, and especially in fourth section of the introduction, where the regional-level gene-environment correlations due to SES-related migration are explained, we clarify the distinction further: “*Selective migration is a form of active gene-environment correlation, where individuals with favourable genetic predispositions can leverage these to improve their environmental circumstances by moving to an economically/socially more favourable region, which in turn improves their outcomes in life. This process has been shown to increase geographic clustering of genes associated with educational attainment and decrease the geographic clustering of PCs that capture older ancestry differences.*”¹⁴ The last sentence of this quote refers to Extended Data Figures 2 & 3 of Abdellaoui et al (2019), where we show that polygenic scores cluster more strongly for current address than for birth place, while for ancestry-informative PCs, it’s the other way around, the PCs cluster more strongly for birth place than for current address.

5. I'd like the authors to further convince the reader that birthplace is truly qualitatively distinct from current address. Aren't the parents' current address at the time of conception potentially influenced by the same genetic variants that are passed on to the child, such that an inherited SNP or genetic predisposition could cause the birthplace in some sense? I'll be honest I find it difficult to think through, and I imagine readers will as well.

We agree with the reviewer that the birth place is likely to be influenced by the same genetic effects that are “affecting” the current address. The geographic clustering of SES is present in both the birth place and the current address, so in that sense, they are not “qualitatively” different. When people move away from their birth place however, they are likely to move to a richer neighbourhood if they can afford it. There are both genetic and environmental explanations for the offspring ending up able to move to an economically more prosperous place than that of their parental residence, for example: 1) the child inherits a higher polygenic score from the combination of the parents than the parents individually have (which is a likely consequence of assortative mating, which is especially strong for SES), and 2) because of a higher education and/or income of the parents, the child may have enjoyed

support through a more favourable environment than the parents had, which may have increased their chances to climb up the social ladder. We now address this in more detail in the third section of the discussion section as follows:

“The reduction in SES-related polygenic signal is more widespread and more substantial and significant when controlling for current address than for birth place. This is in line with increased geographic clustering of complex trait variation for current address (and decreased geographic clustering of ancestry-related variation)¹⁴ due to SES-related migration, adding active on top of passive gene-environment correlations. The parents’ address is likely influenced by the same genetic variants that they passed on to the offspring, and there are a multitude of mechanisms that could lead to a situation in which the offspring ends up increasing the already significant gene-environment correlation through their own migration in adulthood. These explanations could be both environmental and genetic, or a combination thereof. Passive wealth transfer from parent to child throughout life could play a role (e.g., financial support or inheritance), and the child could potentially have inherited a higher polygenic score from the combination of the parents than the parents individually have, which could happen as a consequence of assortative mating, known to be especially strong for SES.^{28,29}”

6. This last comment is simply food for thought, I am not suggesting the authors necessarily take this on. The authors are attempting to understand gene-environment correlation between geographical location (as a proxy for whatever) and a wide range of phenotypes. Can you use migration information (e.g., the change between birth and current location) to help disentangle the relative roles of genes and environment affecting the development of a phenotype? Surprisingly, on a brief literature search I was unable to find such a study in the UK Biobank.

We agree with the reviewer that this is indeed interesting food for thought, and that it could indeed be possible to extract more information about the relative role of genes versus environment from migration histories. We also agree that it would probably be better to do this in a stand-alone study, as our article is already quite comprehensive and such an analysis would require much space because it deserves to be elaborated in detail. We did suggest this for future research now in the discussion in the segment below (the last sentence in bold):

“The parents’ address is likely influenced by the same genetic variants that they passed on to the offspring, and there are a multitude of mechanisms that could lead to a situation in which the offspring ends up increasing the already significant gene-environment correlation through their own

*migration in adulthood. These explanation could be both environmental and genetic, or a combination thereof. Passive wealth transfer from parent to child throughout life could play a role (e.g., financial support or inheritance), and the child could potentially have inherited a higher polygenic score from the combination of the parents than the parents individually have, which could happen as a consequence of assortative mating, known to be especially strong for SES.^{28,29} **More complete information about migration history in combination with parental and offspring genotypes could potentially aid in more accurate disentanglement of (indirect) genetic versus environmental influences.***

Reviewer #2:

Remarks to the Author:

Abdellaoui et al. investigate and interpret the impact of correcting for geographic region (e.g. current address), a genetically heritable trait, in GWAS of educational attainment and related traits.

Major comments:

1. I disagree with the interpretation of "geographic confounding" that is provided throughout this paper. Although this is more of a disagreement with the interpretation than a disagreement with the scientific analyses, it still represents a fundamental disagreement.

In detail, the observation that geographic region is a genetically heritable trait (after correcting for genetic ancestry) was made by Abdellaoui 2019 NHB, which is appropriately cited. The interpretation of Abdellaoui 2019 NHB is that migration is partly driven by SES, which is genetically heritable. Notably, that paper did not ever use the word "confounding". Instead, it simply stated that geographic region is a genetically heritable trait. Thus, it seems to me that the current paper is partitioning the genetically heritable component of educational attainment into the part that can be explained by geographic region and the part that cannot be explained by geographic region. This is analogous to partitioning the genetically heritable component of educational attainment into the part that can be explained by cognitive ability and the part that cannot be explained by cognitive ability (Demange 2021 NG; same last author as this paper); that paper also did not ever use the word "confounding". The same analogy applies to the example on p.3 of the current paper involving a scenario of more expensive insulin leading to a genetic correlation between educational attainment and untreated diabetes. In that case, one could partition the genetically heritable component of untreated diabetes into the part that can be explained by educational attainment and the part that cannot be explained by educational attainment.

I also disagree with the term "gene-environment correlation", because geographic region is not a solely environmental variable, but rather a genetically heritable trait, as shown by Abdellaoui 2019 NHB. Indeed, the demonstration in the current paper that current address is more important than birthplace provides compelling proof that geographic region is genuinely influenced by the target

14individual's genetics, as opposed to genetic ancestry or parental genotype effects. Thus, it is important to either (a) agree with me and drop the use of the term "confounding" throughout the paper, or (b) disagree with me but include a balanced presentation of my point of view in the Introduction and Discussion sections, while providing a compelling explanation of why my point of view is not preferred. Would NG readers be interested to see another partitioning of the genetically heritable component of educational attainment, beyond that previously provided by Demange 2021 NG? I suggest that the answer could potentially be yes, as this partitioning of the genetically heritable component of educational attainment is completely different and unrelated to that provided by Demange 2021 NG, and geographic region is an interesting genetically heritable trait.

We would like to thank the reviewer for their valuable comments and suggestions, which have significantly improved the manuscript. This reviewer's first major, and important, comment (interpretation of geographic confounding and dropping the term 'confounding') has had a substantial impact on our manuscript. We have significantly changed and expanded the manuscript thanks to the valid points raised by Reviewer 2. We changed the title of the manuscript, do no longer refer to the processes we observe as "confounding", and we have added elaborate within-family polygenic score analyses to investigate the nature of the gene-environment correlations. We thank the reviewer for improving our manuscript with this feedback.

In addressing this comment, we'd like to start with saying that we agree with the final sentence of this comment "*I suggest that the answer could potentially be yes, as this partitioning of the genetically heritable component of educational attainment is completely different and unrelated to that provided by Demange 2021 NG, and geographic region is an interesting genetically heritable trait.*" It is indeed different than the partitioning done in Demange et al, 2021 and more than just an interesting genetically heritable trait. We see the polygenic signal of educational attainment as a bundle of underlying traits that make it easier or harder to do well in school, and cognition is one of the more important traits herein (Demange et al, 2021). The reviewer suggests that geographical region might be considered as just one of those underlying polygenic traits, we view that differently, as geographic region does not add a different polygenic signal to the trait, it just inflates the existing one (see our reply to comment 2). A substantial part of the history of complex traits genetics has focused on disentangling genetic and environmental effects from each other (e.g., many decades of twin studies). We now see in the GWAS era that these genetic and environmental effects can be intertwined and more difficult to separate from each other. We think that we should be careful before we label an environmental component as genetic, because that may have downstream consequences to the discourse that are hard to oversee. For example, it has been reported several times (including Abdellaoui et al, 2019) that fast-food restaurants are more prevalent in lower SES regions. We think

it would make more sense to call the fact that fast-food restaurants are correlated with SES-related genetic variation gene-environment correlations than to call the number of fast-food restaurants in your area a heritable trait. While the geographic location may indeed be influenced by genetic effects, the environmental influences that you get exposed to in that geographic region should be referred to as “environment” (that is what the “environment” in “gene-environment correlation” refers to). We do agree with the reviewer that labelling this component as “confounding” is probably also not helpful, so we therefore chose to stick with the more accurate label of “gene-environment correlation”.

A major change we made due to the reviewer’s comment is that we have now updated the title and the entire manuscript to not refer to the effects we observe as confounding, but as part of the pathways in between genetic variation and the outcome. We have also expanded the article to further investigate the nature of these pathways by including educational attainment polygenic score analyses in siblings. These analyses were performed to detect whether regional-level gene-environment correlations result in an additional inflation beyond the already known inflation that occurs due to gene-environment correlations on a family-level (Selzam et al, 2018; *Comparing Within- and Between-Family Polygenic Score Prediction, AJHG*), and whether these include additional *active* gene-environment correlations.

Furthermore, the reviewer says “*Indeed, the demonstration in the current paper that current address is more important than birthplace provides compelling proof that geographic region is genuinely influenced by the target individual’s genetics, as opposed to genetic ancestry or parental genotype effects.*”. The reason that current address is more important does not mean that birth place is not an important part of the process. The birth place already contains a substantial part of the gene-environment correlation, which we now address in the discussion as follows:

“The reduction in SES-related polygenic signal is more widespread and more substantial and significant when controlling for current address than for birth place. This is in line with increased geographic clustering of complex trait variation for current address (and decreased geographic clustering of ancestry-related variation)¹⁴ due to SES-related migration, adding active on top of passive gene-environment correlations. The parents’ address is likely influenced by the same genetic variants that they passed on to the offspring, and there are a multitude of mechanisms that could lead to a situation in which the offspring ends up increasing the already significant gene-environment correlation through their own migration in adulthood. These explanation could be both environmental and genetic, or a combination thereof. Passive wealth transfer from parent to child throughout life could play a role (e.g., financial support or inheritance), and the child could potentially have inherited a higher polygenic score from the combination of the parents than the

parents individually have, which could happen as a consequence of assortative mating, known to be especially strong for SES.^{28,29}

2. Demange 2021 NG computed and analyzed summary statistics for the non-cognitive part of educational attainment as a linear combination of (i) summary statistics for educational attainment and (ii) cognitive ability. Would it likewise be possible in the current paper to compute and analyze summary statistics for the part of educational attainment that can be explained by geographic region as a linear combination of summary statistics for (i) educational attainment and (ii) the part of educational attainment that cannot be explained by geographic region (both of which are analyzed in the current paper)? It seems to me that the answer should be yes, or perhaps this partitioning could even be done at the level of individual-level data. This is an important question that must be addressed in the paper. Specifically, either the part of educational attainment that can be explained by geographic region should be included in the SNP-based heritability and genetic correlation analyses, or the paper should provide a compelling explanation as to why this is not possible.

We agree with the reviewer that this would be a very sensible analysis to perform, to subtract the geography-corrected educational attainment signals from the uncorrected educational attainment signal in order to analyse the part that can be explained by geographic region. However, the genetic correlation between the geography-corrected and the uncorrected signal is .99 (SE = .002) for educational attainment. When the genetic correlation is that high, a GWAS-by-subtraction does not work, because you would remove all of the polygenic signal. The extremely high genetic correlations also show that there is not a qualitatively different genetic component that is being subtracted from educational attainment by controlling for geography.

3. What would happen if you corrected only for latitude and longitude of current address and/or birthplace, instead of dummy variables representing 859-1,959 distinct regions. Would you get similar results? I suggest that this is an important analysis that should be performed. If it would turn out that correcting for latitude and longitude would be sufficient to produce similar results (although I doubt this), that would be far simpler and thus preferred in future research projects.

We thank the reviewer for this suggestion. We have performed this analysis, and the results are very different from controlling for geographic local authority or MSOA region. Controlling for latitude and longitude reduced the heritability for very different traits, namely anthropomorphic and cardiovascular traits. The SNP-based heritability was most strongly affected for height, which was

reduced from 41% to 13%. Other significant reductions were observed for hip circumference, mean corpuscular volume, and mean corpuscular haemoglobin. This effect could not be explained by a reduction in population stratification as captured by the LD score intercept, which increased after controlling for latitude and longitude. We are not sure what could explain the very strong drop in heritability for height after controlling for latitude and longitude. One potential explanation could be that, since latitude and longitude correlate strongly with the PCs, the reduction was caused by multicollinearity. We summarized the results in Supplementary Figures 2 and 3, Supplementary Tables 2 and 3, and in the results section as follows:

“Controlling for assessment centre did not significantly reduce any of the heritability estimates, while controlling for latitude and longitude did, but for very different traits, namely anthropomorphic and cardiovascular traits (see Supplementary Figures 2 and 3). The SNP-based heritability was most strongly affected for height, which was reduced from 41% to 13% (for both birth place and current address separately and combined). Other significant reductions were observed for hip circumference, mean corpuscular volume, and mean corpuscular haemoglobin. This effect could not be explained by a reduction in population stratification, as the LD score intercept was increased after controlling for latitude and longitude (Supplementary Table 2). This reduction in heritability only occurred when controlling for 100 PCs and latitude and longitude combined, but not when controlling for only PCs or only longitude and latitude (Supplementary Table 2); since latitude and longitude correlate strongly with several PCs, this effect may have been caused by multicollinearity.”

4. What would happen if you corrected for dummy variables representing a much smaller number of distinct regions (e.g. 10 regions, instead of 859-1,959 distinct regions). Would you get similar results? I suggest that this is an important analysis that should be performed. If it would turn out that correcting for dummy variables representing a much smaller number of distinct regions would be sufficient to produce similar results (although I doubt this), that would be far simpler and thus preferred in future research projects.

We agree that it is worth investigating whether future research could also use larger regions to get similar or better results. We have therefore now expanded the article to also include the substantially larger local authority regions to the analyses (see Figures 3-6 and the entire section “*Part 2: GWASs Controlled for Geography*” of the results section). The smaller MSOA regions perform better overall than the larger local authority regions in terms of reducing gene-environment correlations in the GWAS signal.

5. In the genetic correlation analysis, I suggest that it would of interest to also include the following traits: (a) educational attainment in non-UK Biobank European samples (see e.g. Demange 2021 NG), (b) the cognitive component of educational attainment and the non-cognitive component of educational attainment (see Demange 2021 NG).

We have added genetic correlations between the cognitive and non-cognitive components; see Supplementary Figures 6 and 7. We see similar changes after controlling for MSOA region, except for a small number of traits that are differentially correlated with cognitive versus non-cognitive skills, which also show differential effects for controlling for geographic region. Interestingly, income shows one of the strongest differences, with little change for the cognitive skills after controlling for MSOA, but a significant decrease of its genetic correlation with non-cognitive skills. We describe this result in the Results section (paragraph *Genetic Correlations*) as follows:

“We also computed genetic correlations between cognitive and non-cognitive components of educational attainment, as extracted by Demange et al (2019)25 (Supplementary Figures 6 and 7). Similar changes occurred after controlling for MSOA region, except for a small number of traits that are differentially correlated with cognitive versus non-cognitive skills, which, accordingly, show differential effects for controlling for geography. Interestingly, income shows a particularly strong difference, with little change for the cognitive skills after controlling for MSOA, but a significant decrease for its genetic correlation with non-cognitive skills.”

Since educational attainment in non-UK participants captures mostly the same polygenic signal, we did not include this, as it would be a comparison of mostly the same polygenic signals.

6. The paper states that correcting for current address has a larger a greater impact than correcting for birthplace. This seems to be a reasonable statement based on the information that is provided (#traits with significant change in SNP-based heritability or genetic correlation, when correcting for birthplace or current address or birthplace + current address). However, as this is an interesting result, a more precise quantification would be preferred. Specifically, in both SNP-based heritability and genetic correlation analyses, for how many traits is the impact of correcting for current address significantly larger than the impact of correcting for birthplace, at an appropriate and explicitly specified significance level?

The question of the additional signal of gene-environment correlation of current address on top of the birth place is now addressed more formally in the polygenic score analyses in siblings, specifically Figures 2A, where the addition of between region and between families is compared, and in Figure

192D, where the results of model 5 are shown. Furthermore, we have now made it visually more clear whether the changes in heritability and genetic correlations in Figures 5 and 6 are significant for birth place, current address, or both, by placing the significance stars at the bottom of the plot with distinct colouring.

Minor comments:

-- p.2: "reflective of a causal effect of that allele": this statement does not account for the widely acknowledged effects of LD-tagging, and should be adjusted accordingly.

Thank you for pointing this out. This part of the introduction has been improved and shortened, and this sentence has been removed.

-- p.2: population stratification is described as an entirely environmental phenomenon. However, it can also be partially or entirely genetic. (For example, one could envision a hypothetical scenario in which all differences in height between human populations are genetically driven. This could still lead to false-positive associations at individual loci that do not causally impact height.) Please modify the text to note this possibility.

We have modified the text accordingly as follows:

*“Genome-wide allele frequency differences between populations due to genetic drift and/or natural selection are detectable even within reasonably homogenous populations.⁶⁻⁸ These older ancestry differences generally show strong correlations with geography,⁶⁻⁹ which could result in correlations with environmental influences that show regional differences. **Population stratification could then lead to biases in GWASs that are not only due to systematic genetic differences, but also additional confounding due to correlated environmental differences between (sub)populations.**”*

-- p.2: population stratification: consider citing/discussing Haworth 2019 NC, which pertains specifically to the UK Biobank data set analyzed in this study.

We agree that this seems a relevant study to cite. However, in a previous paper, we have argued that the geographic structure addressed in this article was likely not interpreted correctly, which is why we chose not to refer to this paper here. In Abdellaoui et al (2019; *Genetic correlates of social*

stratification in Great Britain, Nature Human Behaviour), we refer to this paper as follows in the paragraph “*Geographic clustering of ancestry and complex trait variation*”:

“It has been argued that geographic clustering of genetic variation related to EA in Great Britain is probably due to (subtle) ancestry differences or ascertainment bias (Haworth et al, 2019). In the Supplementary Information, we discuss in more detail why these are unlikely to drive our observations (see the section ‘Population stratification and ascertainment bias’). Instead, we explored a more likely explanation; namely, recent SES-related migrations.”

-- p.3: "geographic confounding has been shown to exist [Abdellaoui 2019 NHB]". This statement is incorrect, as the paper of Abdellaoui 2019 NHB does not include the word "confounding". See Major comment 1. Please modify the text accordingly.

We agree with the reviewer and have now refrained from referring to our observations as confounding throughout the manuscript, and instead refer to these effects as gene-environment correlations (see our comments on Major comment 2).

-- p.3: I suggest that it would be good to clarify here why only a subset of 271,457 of the 456,064 European-ancestry samples were included. Was it primarily because latitude and longitude was available for only a subset of European-ancestry samples? Or primarily because discarding MSA regions with <100 samples caused many samples to be excluded?

The subset was chosen to include only individuals that could be mapped to all geographic variables used in our study, in order to make comparisons between the analyses possible. We have now clarified this at the beginning of the results section in the section *Data and Analysis*:

“In the second part, we run linear-mixed model (LMM) GWASs on 56 complex traits measured in up to 254,387 UK Biobank participants of European descent. Only individuals that could be mapped to all geographic variables were included, in order to make comparisons between the analyses possible.”

-- p.4: "The LMM GWAS controls for cryptic relatedness and population stratification by including a genetic relatedness matrix (GRM) in the model [Jiang 2019 NG]". This statement is incorrect. In the method of Jiang 2019 NG, PCs are used to correct for population stratification, and a sparse GRM is used to correct for cryptic relatedness. Because sparsity is imposed on the GRM, the GRM does not correct for population stratification. (The word "additional" in the following sentence is also incorrect.) Please modify the text accordingly.

We thank the reviewer for noting this and we have now modified the text as follows:

Results section: "*The LMM GWASs controls for cryptic relatedness by including a genetic relatedness matrix (GRM) in the model.*"

Methods: "*We created genetic relatedness matrices (GRMs) to include in our LMM GWAS in order to control for cryptic relatedness.*"

-- p.4: "see Supplementary Table 1 for full list of traits and sample sizes": the NG submission included two supplementary table files, one with SNP-based heritability estimates and one with genetic correlation estimates. Neither of these files included sample size information, which should be included.

We understand that this was unclear. We added the Supplementary Table that contains the sample sizes at the end of the article with the Supplementary Figures and Tables. The files that the reviewer refers to are now referred to in the manuscript as "Supplementary Excel File" as opposed to "Supplementary Table" and only contain heritability estimates and genetic correlations. We hope this will make it more clear for the reader when they navigate to the Supplementary Information on the Nature Genetics website, where they will find a separate document entitled "Supplementary Materials" that includes all the Supplementary Figures and Supplementary Tables that can fit in a pdf (including the sample sizes) and a separate Excel file with all the heritability estimates and genetic correlations.

-- p.5: "significant decrease": please state in this paragraph the criteria for significance of the decrease that is employed in this paragraph.

We describe how the changes in heritability and genetic correlations are derived in more detail now in the methods section, in the paragraph “*SNP-Based Heritability and Genetic Correlation*”:

“The standard error of the difference between the heritability and genetic correlations based on the different specifications cannot easily be estimated directly, because the GWASs for which we want to obtain the differences are based on the exact same sample, and their standard errors are therefore highly dependent. Therefore, we estimated the standard error of the differences in heritability and genetic correlations with Genomic SEM, which allows us to account for the dependence between the estimates of the SNP-based heritability and genetic correlations. Genomic SEM accounts for this dependence by explicitly considering the correlation between estimates of heritability and/or genetic correlations that are modelled jointly (for details see Grotzinger et al 2019)²⁴. To obtain the p-value of the difference, we divide the estimate of the difference by its standard error and perform a Z-test. Significance is then determined after FDR-correction of the p-value.”

-- p.5: what is the interpretation for these decreases? It may be preferred to provide some interpretation of the results in the Results section, instead of exclusively in the Discussion section.

We have now added an interpretation for these results in the Results section in addition to the Discussion:

*“The largest number of traits with a significant reduction in SNP-based heritability was observed after controlling for MSOA regions based on both birth place and current address: household income (from 7% to 4%), educational attainment (from 14% to 9%), age at first birth (from 14% to 9%), time spent watching television (from 12% to 9%), overall health (from 8% to 7%), and fluid intelligence (from 22% to 19%). **These traits have a strong relationship with SES, especially income and educational attainment, which showed the strongest reduction and the strongest geographic clustering (Supplementary Figure 1), making it likely that their polygenic signal was inflated due to gene-environment correlations.**”*

-- p.6: "Genes associated with socio-economic success can have an influence on the neighborhood that people can afford to live in": although I find this to be a compelling statement (and a further affirmation of my Major comment 1), this statement is not specific to genetic correlations, and thus I suggest that it should be moved to earlier than the genetic correlations section, which should

focus on genetic correlations.

We believe that the second part of the sentence that follows “*The environmental exposures that differ between neighbourhoods and regions can have effects on a wide range of physical and mental health outcomes, which can cause genes that are associated with socio-economic success to also become associated with these physical and mental health outcomes*” is essential to the interpretation of the genetic correlation analyses. Since these two sentence belong together, we have chosen to leave this sentence in. We hope the reviewer agrees with our choice to introduce the genetic correlation paragraph in this way. It also helps the reader interpret the changes in genetic correlation that we describe in the Results section, in line with the reviewer’s previous comment and their second comment below.

-- p.6: "significant change (FDR corrected)": please state in this paragraph the precise significance threshold.

In text, we now instead refer to the Figures, and in all Figures we state the significance threshold as follows: “*FDR-corrected p-values < 0.05*”.

-- p.7: what is the interpretation of the significantly weaker genetic correlations? It may be preferred to provide some interpretation of the results in the Results section, instead of exclusively in the Discussion section.

The interpretation of a change in genetic correlations are explained in the results section as follows:

“Genes associated with socio-economic success can have an influence on the neighbourhood that people can afford to live in, and thus on the quality of people’s living environment. The environmental exposures that differ between neighbourhoods and regions can have effects on a wide range of physical and mental health outcomes, which can cause genes that are associated with socio-economic success to also become associated with these physical and mental health outcomes. We investigated whether controlling for regional differences decreases genetic correlations with SES by comparing genetic correlations before and after correcting for geographic region.”

Because of the increased length of the article, especially with the new additions in the revision, we could not afford to elaborate much more on the interpretations in the results section. We agree with the reviewer that it would have been nice to do so if we had more room, but the discussion is generally where more elaborate interpretations of the results can be given, and especially in our case, due to the length of our article, we chose to discuss most of the interpretations and implications in the discussion.

-- p.7: I disagree with the statement that within-family analyses would remove the part (of the genetically heritable component of educational attainment) that is due to geographic region. Note that current address, which is shown in this paper to be more important than birthplace, differs among adult siblings. Either this statement should be removed, or this point of disagreement with this statement should be discussed.

We have now addressed this in great detail in the polygenic score analyses in siblings. We now describe in the results section, paragraph “*Part 1: Polygenic Scores in Siblings*” how between-family variation does indeed capture a substantial portion of the geographic region effect (model 4), and that the current address that the siblings do not share explain an additional amount of gene-environment correlation (Figures 2A and 2D).

-- p.15: For the genetic correlations with educational attainment that were adjusted for geographic region, were these estimated by analyzing summary statistics for (i) educational attainment adjusted for geographic region and (ii) the second trait, or by analyzing summary statistics for (i) educational attainment adjusted for geographic region and (ii) the secondary trait adjusted for geographic region? I am guessing that this distinction would make little difference, but it is still appropriate to be clear about exactly which analysis was performed.

We have now clarified in the section referred to by the reviewer (the paragraph *SNP-Based Heritability and Genetic Correlation* in the methods section) that we computed the genetic correlation for i) educational attainment adjusted for geographic region and (ii) the secondary trait adjusted for geographic region as follows: “*For geography-corrected traits, genetic correlations were computed with either educational attainment or income that themselves were also corrected for the same geographic variable.*”

-- Figure 2: consider restricting to a subset of 10 or fewer traits (it is not mandatory for main Figures to include all results of all analyses, which can be provided in Supp Tables), and labeling the most interesting points in panels A-C by color-coded trait name, rather than trait category.

We agree with the reviewer that the visibility of the traits in this Figure was not optimal. We have shortened several trait names and increased the font of the traits in all Figures, and we have now labelled the significant traits in all scatter plots that showed all 56 traits in all Figures. We hope the reviewer agrees with us leaving in all 56 traits after making them more readable, as this would make it easier for the reader to oversee the overall impact of our analyses without having to navigate to the Supplementary Materials.

-- Figure 3: consider restricting to a subset of 10 or fewer traits (it is not mandatory for main Figures to include all results of all analyses, which can be provided in Supp Tables).

Please see our reply to the previous comment.

-- Figure 4: while I agree that within-family GWAS are less powerful at current sample sizes, I find this to be an obvious point that is more deserving of a Supp Figure than a main Figure for visualization.

Besides showing that within-family GWASs are less powerful at current sample sizes, as displayed by the large standard errors, this Figure also visualizes the decrease in heritability estimates of educational attainment when signal due to gene-environment correlation is accounted for with different approaches. Nevertheless, we agree with the reviewer that this Figure would perhaps fit better as a Supplementary Figure, so now we have made it Supplementary Figure 8.

-- Data availability: the authors should publicly release summary statistics for educational attainment and household income, both uncorrected and corrected for current address and/or birthplace. Public release of summary statistics is definitely permitted by UK Biobank.

When the article is published, we will release all of the GWAS summary statistics that were produced will be made available at a public repository, and we will also share these data with UK Biobank.

-- Data availability: it would be even better if individual-level data for current address and/or birthplace (859-1,959 distinct regions as inferred in this paper) could be released through UK Biobank, although I am unsure about the mechanism for doing so.

We will share all these data with UK Biobank when the article is published, so researchers can apply for access through UK Biobank.

Decision Letter, first revision:

7th Mar 2022

Dear Dr Abdellaoui,

Your Article, "Regional-level gene-environment correlations affect genome-wide association studies" has now been seen by 2 referees. You will see from their comments below that while they find your work of interest, some important points are raised. We are interested in the possibility of publishing your study in Nature Genetics, but would like to consider your response to these concerns in the form of a revised manuscript before we make a final decision on publication.

We therefore invite you to revise your manuscript taking into account all reviewer and editor comments. Please highlight all changes in the manuscript text file. At this stage we will need you to upload a copy of the manuscript in MS Word .docx or similar editable format.

*2) If you have not done so already please begin to revise your manuscript so that it conforms to our Article format instructions, available [here](http://www.nature.com/ng/authors/article_types/index.html). Refer also to any guidelines provided in this letter.

27*3) Include a revised version of any required Reporting Summary:

[REDACTED]

We hope to receive your revised manuscript within four to eight weeks. If you cannot send it within this time, please let us know.

Sincerely,
Wei

Wei Li, PhD
Senior Editor
Nature Genetics
New York, NY 10004, USA
www.nature.com/ng

Reviewers' Comments:

Reviewer #1:

Remarks to the Author:

This is my second review of this manuscript. The purpose of the research would appear to have changed quite substantially. The prior manuscript was heavily focused on understanding potential confounding effects of geography, while the current manuscript is focused on active G-E correlation. I apologize in advance that the below therefore reads more like an initial review, versus a second review.

Overarching comments

#####

The nature, origins, and extent of passive and active gene-environment correlation would seem to be the central question of this manuscript and could represent a significant contribution to the scientific literature. G-E correlation remains a terribly difficult and important problem to tackle, and I am delighted that this team is taking this question by the horns. While definitive answers may be elusive, the paper offers a few tantalizing clues. For example, under an additive genetic model, comparisons of within-family to between-family effects are one way to understand and correct for passive G-E correlation. This has been done before in multiple reports, but the contribution here remains helpful. The authors then take these issues to the next level, and focus on the more difficult question of whether one may compare within-region to between-region effects to further control for active (and evocative?) G-E correlation.

It is important not to lose sight of the importance of the problem of G-E correlation, and the need for research efforts to address it. The present paper is an interesting attempt to do this, and the authors should be commended for an interesting and thought-provoking piece of work. As a dutiful reviewer, I also have a critique that I hope is helpful for the authors and editor to consider, which I lay out below. On the issue of active G-E correlation, the authors show that the UK Biobank genetic data are consistent with the fact that education and income are heritable and geographically stratified. This stratification increases as kids grow up, become mobile, develop their career, and build assets. To the extent that these behaviors are heritable, statistical controls for thousands of geographic locations may reduce the heritability of these phenotypes as genetic variance is controlled.

The authors observe reductions in heritability for a few outcomes but how to interpret this result, and therefore what to do about it in GWAS, is complicated in the opinion of this reviewer. The issue is illustrated in the untreated diabetes example provided in the manuscript. A defensible interpretation of the hypothetical example is that it does not describe a spurious (or "inflated" or "biased") genetic correlation induced by the policy shift, but rather a new environmental context in which there is a real genetic effect of education-increasing alleles on diabetes outcomes. That is, the hypothetical policy change shifts the causal framework underlying diabetes to include a larger relative importance of

29education. If one is conducting a GWAS of diabetes outcomes, should one really control for education (or geography as its proxy) in this instance? I'm not sure how the authors would answer this question. It probably depends on the scientific or practical issues at hand. If one did implement such a control, a legitimate argument could be made that they would erroneously "correct" for a crucial source of variance. Indeed, from a prevention/intervention standpoint, education or a revert of the policy may be the most actionable source of genetic variance. The overarching issue here reminds me of a classic article on SES and schizophrenia (Meehl [1971]. High school yearbooks: A reply to Schwarz. *Journal of Abnormal Psychology*, 77, 143-148.)

If any of the above is correct with respect to heritability estimates, then it is less clear how to interpret the genetic correlation results. The authors speculate reasonably about various observed changes in genetic correlations in the discussion. But without more specific hypotheses or expectations laid out in the introduction, the explanations could be interpreted by some readers as ad hoc, even though that is of course not the authors' intent.

A remaining concern from my last review is to determine what is being proxied by the geographic location. The MSOAs are defined by demographic factors like wealth and education, potentially leading to circular logic in the present analyses. What the geography proxies, combined with a causal model of the given disease/outcome, would inform whether geographic location should be controlled for in a given study of a given disease.

Copy edit suggestions

#####

The distinction between gene and environment, and gene-environment correlation, could be clearer in some places. For example, in lines 92-94 it is implied that the regional effects are "environmental", despite an introduction that would appear to emphasize the interplay of gene and environment on complex traits, and past work indicating geographic location is heritable. The present design would not seem capable of determining whether a given regional effect on a complex trait is environmental. This happens elsewhere in the manuscript, including line 346 where it is implied that the geographical region has an "environmental" impact. A careful read throughout the paper focused on the use of these terms may help improve clarity.

Line 240 – can you conclude that the polygenic signal was "inflated"? To me, that implies that the prior estimate was wrong and the current estimate is more right. I would think instead of "inflated" one could say "larger because it captured effects attributable to gene-environment correlation, which may be genetic or environmental or both in nature". In fact, one might argue the current estimate is downwardly biased because it does not include important sources of variance, such as active G-E.

Line 27 – GWAS not necessarily just for common variants. Consider removing word "common".

Line 66 – selective migration does not need to act on genotype, but only on phenotype. That is, the favorable predispositions do not need to be heritable for migration to be selective.

Line 19 in the abstract – does the social stratification introduce g-e correlations? Or does the heritable phenotype introduce the social stratification? Or both? Perhaps all you can say is that a G-E correlation exists, and appears to be active G-E.

Line 568-569 - I don't follow the equation. You're summing over 1-N, but I don't know what N is (I assume it's the number of regions), and k is not indexed in the summations. Should the first term after the sum be PRS_{ijk} ?

Line 576 – Can you describe in some more detail how this bootstrap procedure breaks the dependence in the models? A separate issue, when resampling I would think you want to resample sibling pairs not individuals, although in my experience in twin studies it doesn't matter much.

Thank you for the opportunity to learn more about this work. I am in the habit of signing my reviews.

Scott Vrieze

Reviewer #2:

Remarks to the Author:

I am satisfied with the responses to my comments, and those of the other reviewer. An all-new section on Polygenic scores in siblings (citing new Figure 1 and new Figure 2) has been added -- this is good content, but it is expected that this would generate additional reviewer feedback.

Minor comments:

p.1 (Title): I find "Regional-level" a bit cryptic, as it is unclear whether this is within or across regions. Perhaps "Gene-environment correlations across geographic regions ..." would work better? This may also help to communicate in the title the specific key fact that genetics is correlated to geographic region.

p.1 (Abstract): would it be possible to state more explicitly in the Abstract the key fact that genetics is correlated to geographic region? "extends these gene-environment correlations to broader geographic regions" sort of hints at this, but perhaps it would be even better to state it explicitly.

p.2 (Introduction): I find "Regional-level" a bit cryptic, as it is unclear whether this is within or across regions. Perhaps "Gene-environment correlations across geographic regions ..." would work better?

p.2-3 (Introduction): consider stating even earlier in this paragraph the key fact that genetics is correlated to geographic region.

p.4: "Gene-environment correlations have been empirically shown to exist on the family-level (ref. 12,16, 20)": please add more detail on the findings of these studies (either here, or in Introduction). Given that ref. 12 refers to parental environment, do these 3 refs refer to same or different phenomena?

31p.4: ref. 20 is the same as ref. 15, please fix.

p.5: Please state more explicitly in the text that you are using PRS for educational attainment to predict traits different from educational attainment. It took me awhile to figure this out.

p.5: Please provide motivation for using PRS for educational attainment to predict traits different from educational attainment (instead of using PRS for educational attainment to predict educational attainment). Also, consider discussing the results of using PRS for educational attainment to predict educational attainment in the Results section (not just in the Discussion section), as this seems quite fundamental.

p.5: I found the description of model 4 to be confusing. Consider changing to: "within-family (within-region), between-family (within-region) and between-region".

p.5: For model 5, I wonder if it might be helpful to include the text "within-family (within-region), between-family (within-region), within-family (between-region) and between-family (between-region)".

p.5: "lower within-family effect in model 2 than in model 4": do you actually mean lower within-family effect in *model 4* than in model 2, e.g. 0.04 for model 4 vs. 0.06 for model 2 for household income in Table 1?

p.5: "lower within-family effect in model 2 than in model 4": do you actually mean lower *absolute* within-family effect in *model 4* than in model 2, e.g. -0.05 for model 4 is lower in absolute size than -0.06 for model 2 for BMI in Table 1?

p.5: Please provide additional explanation for "lower within-family effect in model 2 than in model 4", or lower *absolute* within-family effect in *model 4* than in model 2, or whatever is actually correct here. I think I have intuition for the latter, but I would still like to see more explanation.

p.5: "became significantly weaker": I suggest to specifically mention model 2 and model 4 in this sentence, which will helpfully link this content to the explanation provided at the end of the previous paragraph.

p.6: "When comparing fixed effect estimates within model 4, the majority of traits show higher between-region than between-family effect (Figure 2C)": please add 1 sentence here briefly interpreting the implications of this finding. Results text should not merely state that analysis x produced numerical result y.

p.6: "When comparing fixed effect estimates within model 4, the majority of traits show higher between-region than between-family effect (Figure 2C)": please add 1 sentence here briefly interpreting the implications of this finding. Results text should not merely state that analysis x produced numerical result y.

p.6 (Figure 2): I found the miniscule text (trait categories, trait names, text axis labels, numerical axis labels) to be extraordinarily difficult to read. Please use much larger font.

p.6 (Figure 2): It may not be necessary to include all 56 traits in the main Figure, as this can be overwhelming.

p.6: Would it be of value to include other comparisons (e.g. model 1 vs. model 2) as Supp Figures with brief summary in the text?

p.10: "inflated": I do not agree with this word, which sounds a lot like "confounding" (removed from the manuscript in many places during the last round of revisions, based on my comments). Please consider using a different word. Perhaps "impacted"?

p.14: the citation to Figure 4 should be changed to a citation to Supp Figure 8, as promised in the response to Reviewer 2 minor comment on Figure 4.

p.14: Figure 7 should be changed to Supp Figure 8, as promised in the response to Reviewer 2 minor comment on Figure 4.

p.14: "can be reduced ... by controlling for region": I agree with this statement, but I still feel that the recommendation is not 100% clear. Please provide a clear recommendation. For example, should every GWAS (for relevant traits) correct for geographic region, where practical to do so? Perhaps it would be good to include a statement of the form "If you do not correct for geographic region, you will identify GWAS SNPs with property X; if you correct for geographic region, you will identify GWAS SNPs with property Y".

p.14-16: the Discussion is very long and could easily be shortened.

p.19 (Methods): please mention "model 1", "model 2", "model 3", "model 4" here, to be consistent with Results section.

p.19 (Methods): please include model 5 here, as this is prominently included in the Results section.

p.19 (Methods): should each J be changed to j in 2. and 4.?

Response to Reviewer 2, Comment 6: the response states that the results of Figure 2A,2D shed light on correcting for current address vs. birthplace. I suggest to specifically mention "current address" and "birthplace" in Results section text citing Figure 2A,2D, and/or Discussion text citing Figure 2A,2D.

Author Rebuttal, first revision:

Reviewer #1:

The nature, origins, and extent of passive and active gene-environment correlation would seem to

33be the central question of this manuscript and could represent a significant contribution to the scientific literature. G-E correlation remains a terribly difficult and important problem to tackle, and I am delighted that this team is taking this question by the horns. While definitive answers may be elusive, the paper offers a few tantalizing clues. For example, under an additive genetic model, comparisons of within-family to between-family effects are one way to understand and correct for passive G-E correlation. This has been done before in multiple reports, but the contribution here remains helpful. The authors then take these issues to the next level, and focus on the more difficult question of whether one may compare within-region to between-region effects to further control for active (and evocative?) G-E correlation.

It is important not to lose sight of the importance of the problem of G-E correlation, and the need for research efforts to address it. The present paper is an interesting attempt to do this, and the authors should be commended for an interesting and thought-provoking piece of work. As a dutiful reviewer, I also have a critique that I hope is helpful for the authors and editor to consider, which I lay out below.

We thank the reviewer for his interest, time, and thoughtful assessment of our manuscript. We agree with his assessment on the importance of G-E correlation, especially given how wide-spread its impact seems to be across such a wide variety of physical and mental health outcomes. We are thankful for the second round of comments of this reviewer, which have again led to an improvement of the manuscript. We address his comments below.

On the issue of active G-E correlation, the authors show that the UK Biobank genetic data are consistent with the fact that education and income are heritable and geographically stratified. This stratification increases as kids grow up, become mobile, develop their career, and build assets. To the extent that these behaviors are heritable, statistical controls for thousands of geographic locations may reduce the heritability of these phenotypes as genetic variance is controlled.

The authors observe reductions in heritability for a few outcomes but how to interpret this result, and therefore what to do about it in GWAS, is complicated in the opinion of this reviewer. The issue is illustrated in the untreated diabetes example provided in the manuscript. A defensible interpretation of the hypothetical example is that it does not describe a spurious (or “inflated” or “biased”) genetic correlation induced by the policy shift, but rather a new environmental context in which there is a real genetic effect of education-increasing alleles on diabetes outcomes. That is, the hypothetical policy change shifts the causal framework underlying diabetes to include a larger relative importance of education. If one is conducting a GWAS of diabetes outcomes, should one really control for education (or geography as its proxy) in this instance? I’m not sure how the

authors would answer this question. It probably depends on the scientific or practical issues at hand.

If one did implement such a control, a legitimate argument could be made that they would erroneously “correct” for a crucial source of variance. Indeed, from a prevention/intervention standpoint, education or a revert of the policy may be the most actionable source of genetic variance. The overarching issue here reminds me of a classic article on SES and schizophrenia (Meehl [1971]. High school yearbooks: A reply to Schwarz. *Journal of Abnormal Psychology*, 77, 143-148.)

We agree with the reviewer that these gene-environment correlations raise the complicated question of what to do about them in a GWAS. We dedicate half of our analyses on showing how to *reduce* these effects, namely by controlling for geographic region, but even this part of the manuscript also serves more to raise awareness about the presence of these gene-environment correlations across many GWASs rather than to tell researchers what to do in their GWASs. We do not yet have an approach for how to *fully* remove genetic effects that travel through the social environment, and we also do not know how desirable it is to do so. There is no straightforward answer to this, because it depends entirely on what knowledge the researcher wants to extract from a GWAS. There are, for example, companies that offer to screen embryos for polygenic scores in IVF treatments, and they claim to select embryos *only* for disease risk, with diabetes being one of the disease outcomes, and not for non-disease traits like intelligence or education. For such an application (which is quite controversial and we would not endorse), you would probably want to minimize polygenic effects on SES-related outcomes like educational attainment and income that become part of the GWAS on diabetes through these gene-environment correlations.

One of the intentions of our work is to stimulate researchers to think more carefully about what they are extracting from a GWAS and make them aware of the complexity of the long road from DNA sequence to complex physical and mental health outcomes. This will hopefully also make the community more aware of the dynamic nature of the “genetic architecture” of many physical and mental health outcomes. We put “genetic architecture” here in quotation mark, because opinions may also differ on whether genetic effects that travel through our dynamic social environment can be considered part of the genetic architecture of a physical health outcome like diabetes. This is an important discussion to have, and we have added the following closing sentence to the opening paragraph of the discussion to underscore this, referring to the excellent suggested paper of Meehl (1971):

Page 14: *“We showed here that effects of gene-environment correlations that extend beyond the family environment can be reduced in GWASs by controlling for the region that participants were born in and/or moved to. **Fully removing polygenic signals that affect the outcome via the social environment will be more challenging, and it depends on the research question of the GWAS whether it would be desirable to exclude this causal chain from the polygenic signal.**”²⁸”*

And we added a sentence to the closing paragraph of the discussion to again underscore this important point:

Page 15-16: *“Our findings suggest that effects estimated in GWASs of many phenotypes are affected by gene-environment correlations, and that these are not entirely attributable to processes that take place within a family, but also more broadly attributable to regional social and political processes that correlate to individuals’ genotypes. **We showed how to significantly reduce these effects in GWAS signals. It should depend on the goals of the research whether that would be desirable, as they result from dynamic social circumstances that are part of a true causal chain in between our DNA and complex mental and physical health outcomes.** If GWASs are to remain central to the study of (non-communicable) disease epidemiology and biology, their statistical models, research designs, and conclusions need to more carefully reflect the reality of the social and geographic structure of society.”*

If any of the above is correct with respect to heritability estimates, then it is less clear how to interpret the genetic correlation results. The authors speculate reasonably about various observed changes in genetic correlations in the discussion. But without more specific hypotheses or expectations laid out in the introduction, the explanations could be interpreted by some readers as ad hoc, even though that is of course not the authors’ intent.

We did indeed have clear expectations regarding the changes in genetic correlations induced by gene-environment correlations and we agree that we should have posed these in the introduction. We have now done this follows:

We added the following sentence to the introduction, page 2: *“This can result in an inflation of effect estimates in GWASs on educational attainment or income that are not detectable through LD score regression analysis, as the indirect genetic effect can be considered a true genetic effect which is mediated by the rearing environment. **This could also lead polygenic effects on educational attainment or income to become mixed with signals of GWASs on physical and mental health outcomes that are influenced by these environmental influences.**”*

And updated the following sentence in the introduction, page 3: “For instance, a policy change that makes insulin more expensive and therefore more difficult to purchase for low income groups, will introduce a correlation between alleles related to SES (e.g., **education and income**) and those related to consequences of untreated diabetes, **or any other physical or mental health outcome that is influenced by such socio-economic factors.**”

A remaining concern from my last review is to determine what is being proxied by the geographic location. The MSOAs are defined by demographic factors like wealth and education, potentially leading to circular logic in the present analyses. What the geography proxies, combined with a causal model of the given disease/outcome, would inform whether geographic location should be controlled for in a given study of a given disease.

The reviewer rightfully notes that the MSOA regions have been chosen to be relatively homogeneous with respect to SES by the Office of National Statistics, for the purpose of collecting population statistics for those who have to plan and allocate resources. We don't think this would lead to a fallacy in our argumentation, because if the gene-environment correlations were not there, we would not have observed the effects that we present in our results. We were rather lucky that these regions were defined this way, because if we would have to define regions to control for SES-related gene-environment correlations, we would have tried to define the regions in the same way. On the other hand, this is probably a result of non-genetic studies having already noticed that it is useful to divide the regions in this way, because of the broad impacts of SES on such a wide variety of outcomes. Furthermore, the Local Authority regions, which we also analyze, were not defined by demographic factors like wealth and education, and also show the same effects. The Local Authority regions were the regional definitions used when we detected these regional differences in SES-related genes for the first time in our article from 2019 (*Abdellaoui et al, 2019, Genetic Correlates of Social Stratification in Great Britain, Nature Human Behaviour*).

The distinction between gene and environment, and gene-environment correlation, could be clearer in some places. For example, in lines 92-94 it is implied that the regional effects are “environmental”, despite an introduction that would appear to emphasize the interplay of gene and environment on complex traits, and past work indicating geographic location is heritable. The present design would not seem capable of determining whether a given regional effect on a complex trait is environmental. This happens elsewhere in the manuscript, including line 346 where it is implied that the geographical region has an “environmental” impact. A careful read

throughout the paper focused on the use of these terms may help improve clarity.

We agree, and have made this distinction more clear in the following segments:

Page 3: *“In these GWASs, we reduce polygenic signals that affect the trait through environmental effects by introducing fixed effects for neighbourhoods that are socially and economically more homogenous than the country as a whole”* now reads *“In these GWASs, we reduced the part of the signal that arises due to gene-environment correlations by introducing fixed effects for neighbourhoods that are socially and economically more homogenous than the country as a whole”*

Page 14: *“This can possibly be explained by people attaining their maximum education well before they migrated to their current address, in which case the regional environmental factors of the current address may not have had as great an impact on educational attainment, making the contribution of passive gene-environment correlations more important for educational attainment relative to active gene-environment correlations”* now reads *“This may be explained by people attaining their maximum education before migrating to their current address, which would make the contribution of passive gene-environment correlations more important for educational attainment than active gene-environment correlations.”*

Page 14: *“The geographic region captures a substantial part of the family-level gene-environment correlation and a significant, but smaller, portion of independent regional-level environmental effects.”* now reads *“The geographic region captures a substantial part of the family-level gene-environment correlation and a significant, but smaller, portion of independent gene-environment correlations across regions.”*

Page 14: *“These additional independent regional-level environmental effects were less significant for educational attainment than for traits like BMI, income, and time spent watching TV, even though these analyses were done with an educational attainment polygenic score”* now reads *“These additional gene-environment correlations across geographic regions were less significant for educational attainment than for traits like BMI, income, and time spent watching TV, even though these analyses were done with an educational attainment polygenic score”*

Line 240 – can you conclude that the polygenic signal was “inflated”? To me, that implies that the prior estimate was wrong and the current estimate is more right. I would think instead of “inflated” one could say “larger because it captured effects attributable to gene-environment correlation, which may be genetic or environmental or both in nature”. In fact, one might argue the current estimate is downwardly biased because it does not include important sources of

variance, such as active G-E.

The reviewer raises a good point. We have changed the sentence as suggested, which now reads:

Page 10: *“These traits have a strong relationship with SES, especially income and educational attainment, which showed the strongest reduction and also the strongest geographic clustering (Supplementary Figure 2), making it likely that their polygenic signal was larger because it captured effects attributable to gene-environment correlations.”*

Line 27 – GWAS not necessarily just for common variants. Consider removing word “common”.

We agree and have now removed the word “common”.

Line 66 – selective migration does not need to act on genotype, but only on phenotype. That is, the favorable predispositions do not need to be heritable for migration to be selective.

We agree, and have now re-written the sentence as follows:

Page 3: *“Selective migration can induce gene-environment correlations when individuals with favourable genetic predispositions are more likely to improve their environmental circumstances by moving to a more favourable region.”*

Line 19 in the abstract – does the social stratification introduce g-e correlations? Or does the heritable phenotype introduce the social stratification? Or both? Perhaps all you can say is that a G-E correlation exists, and appears to be active G-E.

Social stratification is defined as the categorization of a society’s people into groups based on socioeconomic factors such as educational attainment and income, which are heritable outcomes. In the society we’re studying (and in many other societies) these socio-economic differences are strongly correlated with geography, with considerable differences in environmental factors between geographic regions, but also in genes. We think it is evident that this entire process - regardless of whether it is driven by the emergent properties of a “society” or by the individuals themselves through their heritable

39phenotypes - is introducing a correlation between certain genetic propensities and environmental differences across geographic regions. We have re-written the closing sentence of the abstract as follows (following also a similar suggestion of reviewer 2 to explicitly state the geographic clustering of DNA in the abstract): *“Our results show that the geographic clustering of DNA and SES introduces gene-environment correlations that affect GWAS results”*

Line 568-569 - I don't follow the equation. You're summing over 1-N, but I don't know what N is (I assume it's the number of regions), and k is not indexed in the summations. Should the first term after the sum be PRS_{ijk} ?

The equation denotes a decomposition of the within-family effect into a within-region and between region effect. The N did indeed denote the number of regions, but have now chosen to denote that term with a bar, as in the rest of the equation, for clarity. The first term after the sum should be PRS_{ij} , but the “within-family” term that is “between-regions” should have a K added to it, which has now been clarified. We thank the reviewer for noting this. The equation now reads:

$$Y_{ijk} = \alpha_0 + \beta_W \left((PRS_{ij} - \overline{PRS_j}) - \overline{(PRS_{ij} - \overline{PRS_j})_K} \right) + \beta_B \overline{(PRS_{ij} - \overline{PRS_j})_K} + \gamma_j + \gamma_k + \varepsilon_{ijk},$$

Line 576 – Can you describe in some more detail how this bootstrap procedure breaks the dependence in the models? A separate issue, when resampling I would think you want to resample sibling pairs not individuals, although in my experience in twin studies it doesn't matter much.

The bootstrap does not break the dependence between the parameters estimated in model 2 and 4, but rather across 1000 repetitions, the dependence between β_{W2} and β_{W4} is reproduced faithfully. Given that the dependence is reproduced, it is then also reproduced in the standard deviation of the differences, allowing us to account for the uncertainty in $\beta_{W2} - \beta_{W4}$ in a way that is sensitive to their correlation.

We clarified this as follows in the methods on page 20: *“Any correlation between β_{W2} and β_{W4} is reproduced faithfully across the bootstrap samples, meaning that the standard error of their differences accounts for this correlation”*.

Reviewer #2:

I am satisfied with the responses to my comments, and those of the other reviewer. An all-new section on Polygenic scores in siblings (citing new Figure 1 and new Figure 2) has been added -- this is good content, but it is expected that this would generate additional reviewer feedback.

We are happy that the reviewer is satisfied and thank them for their thoughtful assessment and constructive comments on the additions to our manuscript. These have again led to an improvement of our manuscript, for which we are very grateful. Please see how we have addressed their comments below.

p.1 (Title): I find "Regional-level" a bit cryptic, as it is unclear whether this is within or across regions. Perhaps "Gene-environment correlations across geographic regions ..." would work better? This may also help to communicate in the title the specific key fact that genetics is correlated to geographic region.

We thank the reviewer for this excellent suggestion. We have changed the title as proposed: "*Gene-environment correlations across geographic regions affect genome-wide association studies*". Furthermore, we have changed *regional-level* throughout the manuscript into *across geographic regions*.

p.1 (Abstract): would it be possible to state more explicitly in the Abstract the key fact that genetics is correlated to geographic region? "extends these gene-environment correlations to broader geographic regions" sort of hints at this, but perhaps it would be even better to state it explicitly.

We stated this fact more explicitly now in the closing sentence of the abstract: "*Our results show that the geographic clustering of DNA and SES introduces gene-environment correlations that affect GWAS results.*"

p.2 (Introduction): I find "Regional-level" a bit cryptic, as it is unclear whether this is within or across regions. Perhaps "Gene-environment correlations across geographic regions ..." would work better?

41We thank the reviewer for this suggestion. We have now changed “regional-level” throughout the manuscript into *across geographic regions*.

p.2-3 (Introduction): consider stating even earlier in this paragraph the key fact that genetics is correlated to geographic region.

We have re-written the first sentence of this paragraph as follows: “*A third source of gene-environment correlations can be found across geographic regions through the geographic clustering of polygenic effects and environmental factors related to regional socio-economic differences*”.

p.4: “Gene-environment correlations have been empirically shown to exist on the family-level (ref. 12,16, 20)”: please add more detail on the findings of these studies (either here, or in Introduction). Given that ref. 12 refers to parental environment, do these 3 refs refer to same or different phenomena?

For reference 12, this is explained in the introduction on page 2 as follows: “*Differences between families in (socio-economic) environmental factors that correlate with genetic factors are also known to affect genetic effect estimates in GWASs. One way in which such correlations arise is if an individual’s outcome has been influenced by parental (rearing) behaviour and socio-economic position through the parental genotype.*¹²”

For reference 20 (which is now 17) this is explained on page 4-5 as follows: “*Polygenic scores for educational attainment have been found to be twice as predictive of education in non-adopted individuals compared with adoptees.*¹⁷”

For reference 16, this is explained on page 5 as follows: “*For educational attainment and IQ, polygenic scores have been shown to explain 60% more variance between families than within families.*¹⁶”

p.4: ref. 20 is the same as ref. 15, please fix.

This has been fixed.

p.5: Please state more explicitly in the text that you are using PRS for educational attainment to predict traits different from educational attainment. It took me awhile to figure this out.

We have now mentioned this in the second sentence of the abstract: *“We first showed with educational attainment polygenic scores in up to 43,516 British adult siblings that genetic effect estimates on SES-related traits capture gene-environment correlations”*.

It is also mentioned in the last paragraph of the introduction: *“To this end, we use educational attainment polygenic scores and phenotypic measures of 56 complex traits in 22,657 adult sibling pairs from UK Biobank.”*

And it is mentioned in the Figure caption of Figure 2: *“Results of sibling educational attainment polygenic score analyses for 56 complex traits.”*

On page 5 in the results section, we explain why we use the educational attainment polygenic score as follows: *“SES-related signals are best captured by GWASs on educational attainment and income. We created a polygenic score for educational attainment rather than income, because educational attainment is the trait with the largest available GWAS that excludes our UK Biobank participants. We used this polygenic score to predict educational attainment as well as 55 other complex traits, as we expected the environmental effects involved in gene-environment correlations to impact a wider variety of outcomes.”*

p.5: Please provide motivation for using PRS for educational attainment to predict traits different from educational attainment (instead of using PRS for educational attainment to predict educational attainment). Also, consider discussing the results of using PRS for educational attainment to predict educational attainment in the Results section (not just in the Discussion section), as this seems quite fundamental.

We have now added the following sentence on page 5: *“We use this polygenic score to predict educational attainment as well as 55 other complex traits, as we expect the environmental effects that become part of the gene-environment correlations to impact a wider variety of outcomes.”*

In addition, we added the result for educational attainment to the five most significant traits in the text on page 6 as well:

*“The five most significant reductions were observed for BMI ($p = 1 \times 10^{-5}$), waist circumference ($p = 1 \times 10^{-4}$), household income ($p = 1 \times 10^{-4}$), time spent watching TV ($p = 3 \times 10^{-4}$), and whole body fat mass ($p = 6 \times 10^{-4}$), possibly because these are traits that are more subject to change after siblings migrated out of the parental residence (see Table 1 for full results for these traits). **Adding the between-region effect to the between-family effect also significantly decreased the individual-level polygenic score effect for educational attainment ($p = 9 \times 10^{-4}$).**”*

p.5: I found the description of model 4 to be confusing. Consider changing to: "within-family (within-region), between-family (within-region) and between-region".

Thank you, we agree that the suggested description is more clear. We have now changed it to:

*“**model 4** includes polygenic scores on the within-family (within-region), between-family (within-region), and between-region as predictors”*

p.5: For model 5, I wonder if it might be helpful to include the text "within-family (within-region), between-family (within-region), within-family (between-region) and between-family (between-region)".

Model 5 only includes within-family (within-region) and within-family (between-region). We have clarified the equation in the methods section to more clearly convey this:

$$Y_{ijk} = \alpha_0 + \beta_W \left((PRS_{ij} - \overline{PRS_j}) - \overline{(PRS_{ij} - \overline{PRS_j})_K} \right) + \beta_B \overline{(PRS_{ij} - \overline{PRS_j})_K} + \gamma_j + \gamma_k + \varepsilon_{ijk},$$

p.5: "lower within-family effect in model 2 than in model 4": do you actually mean lower within-family effect in *model 4* than in model 2, e.g. 0.04 for model 4 vs. 0.06 for model 2 for household income in Table 1?

p.5: "lower within-family effect in model 2 than in model 4": do you actually mean lower *absolute* within-family effect in *model 4* than in model 2, e.g. -0.05 for model 4 is lower in absolute size than -0.06 for model 2 for BMI in Table 1?

Yes, that is indeed what we mean. We are very grateful that this sharp referee caught this error, which has now been fixed:

“An important indicator for the presence of gene-environment correlations beyond the family-level across geographic regions is a weaker within-family effect in model 4 than in model 2”

p.5: Please provide additional explanation for "lower within-family effect in model 2 than in model 4", or lower *absolute* within-family effect in *model 4* than in model 2, or whatever is actually correct here. I think I have intuition for the latter, but I would still like to see more explanation.

We agree that a clarification here would be an improvement, so we have now added the following explanation:

“An important indicator for the presence of gene-environment correlations beyond the family-level across geographic regions is a weaker within-family effect in model 4 than in model 2, since that would imply that a portion of the effect captured on the individual-level, and not by the family, is due to gene-environment correlations across geographic regions”

p.5: "became significantly weaker": I suggest to specifically mention model 2 and model 4 in this sentence, which will helpfully link this content to the explanation provided at the end of the previous paragraph.

This has now been added as follows:

“For ten traits, the individual-level effect of the polygenic score became significantly weaker after adding a between-region effect to the between-family effect (i.e., weaker in model 4 compared to model 2, based on p-values of the difference in effect sizes based on 1000 bootstraps; Figure 2A).”

p.6: "When comparing fixed effect estimates within model 4, the majority of traits show higher

between-region than between-family effect (Figure 2C)": please add 1 sentence here briefly interpreting the implications of this finding. Results text should not merely state that analysis x produced numerical result y.

We have now expanded this sentence as follows:

*"When comparing fixed effect estimates within model 4, the majority of traits show higher between-region than between-family effect (Figure 2C), **implying additional explanatory power of geographic regions not captured at the individual or family level.**"*

We also expanded the sentence that followed:

*"After decomposing the within-family effect into within- and between-region effects (model 5), the majority of traits show higher between-region than within-region effect (Figure 2D), **implying wide-spread active gene-environment correlations.**"*

p.6 (Figure 2): I found the miniscule text (trait categories, trait names, text axis labels, numerical axis labels) to be extraordinarily difficult to read. Please use much larger font.

p.6 (Figure 2): It may not be necessary to include all 56 traits in the main Figure, as this can be overwhelming.

We have now increased the readability of Figure 2 as suggested.

p.6: Would it be of value to include other comparisons (e.g. model 1 vs. model 2) as Supp Figures with brief summary in the text?

We have added two comparisons as Supplementary Figures 1A and 1B. The brief summary in the text reads as follows:

Page 5: *“As expected from Selzam et al¹⁶, individual-level effect decreased after adding between-family effects (model 1 versus model 2, Supplementary Figure 1A).”*

Page 6: *“Part of the between-family effect in model 2 can be explained by between-region effects, reflected by a significant decrease of the between-family effect in the majority of traits after adding the between-region effect (38 traits showed a significant decrease of between-family effects in model 4 compared to model 2, based on FDR-corrected p-values of the difference in effect sizes based on 1000 bootstraps; Supplementary Figure 1B).”*

p.10: "inflated": I do not agree with this word, which sounds a lot like "confounding" (removed from the manuscript in many places during the last round of revisions, based on my comments). Please consider using a different word. Perhaps "impacted"?

We have now rephrased this sentence on page 10 as follows:

“These traits have a strong relationship with SES, especially income and educational attainment, which showed the strongest reduction and also the strongest geographic clustering (Supplementary Figure 1), making it likely that their polygenic signal was larger because it captured effects attributable to gene-environment correlations.”

p.14: the citation to Figure 4 should be changed to a citation to Supp Figure 8, as promised in the response to Reviewer 2 minor comment on Figure 4.

The Figure reference on page 14 was already to Supplementary Figure 8, assuming the reviewer meant this sentence (we can find no other reference to the old Figure 4):

“In addition, while family datasets are growing, large sample sizes of genotyped families are harder to attain, which results in less powerful within-family GWASs with noisier estimates of genetic effects (Supplementary Figure 8).”

p.14: Figure 7 should be changed to Supp Figure 8, as promised in the response to Reviewer 2 minor comment on Figure 4.

We can find no other reference to the old Figure 7, only to Supplementary Figure 8 (see previous comment). We double checked all the references to all Figures and these should be now correct.

p.14: "can be reduced ... by controlling for region": I agree with this statement, but I still feel that the recommendation is not 100% clear. Please provide a clear recommendation. For example, should every GWAS (for relevant traits) correct for geographic region, where practical to do so? Perhaps it would be good to include a statement of the form "If you do not correct for geographic region, you will identify GWAS SNPs with property X; if you correct for geographic region, you will identify GWAS SNPs with property Y".

We have extended this section following a very similar request from Reviewer 1 (see our response on page 2 of this document). This now reads as follows:

"We showed here that effects of gene-environment correlations that extend beyond the family environment can be reduced in GWASs by controlling for the region that participants were born in and/or moved to. Fully removing polygenic signals that affect the outcome via the social environment will be more challenging, and it depends on the research question of the GWAS whether it would be desirable to exclude this causal chain from the polygenic signal.²⁸"

We also emphasized this point in the concluding paragraph of our paper:

"We showed how to significantly reduce these effects in GWAS signals. It should depend on the goals of the research whether that would be desirable, as they result from dynamic social circumstances that are part of a true causal chain in between our DNA and complex mental and physical health outcomes."

p.14-16: the Discussion is very long and could easily be shortened.

We were indeed able to substantially shorten the discussion from 1527 words to 1293 words without losing any of the information conveyed.

p.19 (Methods): please mention "model 1", "model 2", "model 3", "model 4" here, to be consistent with Results section.

We have mention “model 1” – “model 5” in the Methods now as well.

p.19 (Methods): please include model 5 here, as this is prominently included in the Results section.

This has now been mentioned in the methods as well.

p.19 (Methods): should each J be changed to j in 2. and 4.?

No, but we should have capitalized the K’s when referring to the averages (i.e., all terms with bars over them), which we have now done.

Response to Reviewer 2, Comment 6: the response states that the results of Figure 2A,2D shed light on correcting for current address vs. birthplace. I suggest to specifically mention "current address" and "birthplace" in Results section text citing Figure 2A,2D, and/or Discussion text citing Figure 2A,2D.

All of the sibling analyses are based on current address (which is mentioned on page 5). Any geographic effect that is not captured by the family effect is assumed to be caused by a deviation of the current address, unique to the sibling, from the birth address, shared by the siblings, as they were born in the same family. We have made an illustration of this design in Figure 1, where we also specifically and prominently mention *current address* and *birth place*.

Decision Letter, second revision:

Our ref: NG-A57213R1

26th May 2022

49Dear Dr. Abdellaoui,

Thank you for submitting your revised manuscript "Gene-environment correlations across geographic regions affect genome-wide association studies" (NG-A57213R1). It has now been seen by the original referees and their comments are below. The reviewers find that the paper has improved in revision, and therefore we'll be happy in principle to publish it in Nature Genetics, pending minor revisions to satisfy the referees' final requests and to comply with our editorial and formatting guidelines.

Sincerely,
Wei

Wei Li, PhD
Senior Editor
Nature Genetics
New York, NY 10004, USA
www.nature.com/ng

Reviewer #2 (Remarks to the Author):

I am satisfied with the responses to my comments, and those of the other reviewer.

I have only one very minor comment: in the last round of review, both Reviewer #1 (line 240 comment) and Reviewer #2 (p.10 comment) advocated against the use of the term "inflated". However, it seems that this term ("inflated", "inflation", "inflates") is still used in some parts of the text (line 54, line 130, line 139, line 140). The authors might consider whether to use a different term in each case. (Optionally, the authors could even add a sentence to the Discussion section explicitly discussing whether or not the impact of gene-environment correlations ought to be labeled as "inflation".)

Final Decision Letter:

50In reply please quote: NG-A57213R2 Abdellaoui

13th Jul 2022

Dear Dr. Abdellaoui,

I am delighted to say that your manuscript "Gene-environment correlations across geographic regions affect genome-wide association studies" has been accepted for publication in an upcoming issue of Nature Genetics.

Your paper will be published online after we receive your corrections and will appear in print in the next available issue. You can find out your date of online publication by contacting the Nature Press Office (press@nature.com) after sending your e-proof corrections. Now is the time to inform your Public Relations or Press Office about your paper, as they might be interested in promoting its publication. This will allow them time to prepare an accurate and satisfactory press release. Include your manuscript tracking number (NG-A57213R2) and the name of the journal, which they will need when they contact our Press Office.

Please note that *Nature Genetics* is a Transformative Journal (TJ). Authors may publish their research with us through the traditional subscription access route or make their paper immediately open access through payment of an article-processing charge (APC). Authors will not be required to make a final decision about access to their article until it has been accepted. [Find out more about Transformative Journals](https://www.springernature.com/gp/open-research/transformative-journals)

Authors may need to take specific actions to achieve [a](https://www.springernature.com/gp/open-research/funding/policy-compliance-)

compliance with funder and institutional open access mandates. If your research is supported by a funder that requires immediate open access (e.g. according to [Plan S principles](https://www.springernature.com/gp/open-research/plan-s-compliance)) then you should select the gold OA route, and we will direct you to the compliant route where possible. For authors selecting the subscription publication route, the journal's standard licensing terms will need to be accepted, including <https://www.nature.com/nature-portfolio/editorial-policies/self-archiving-and-license-to-publish>. Those licensing terms will supersede any other terms that the author or any third party may assert apply to any version of the manuscript.

Please note that Nature Portfolio offers an immediate open access option only for papers that were first submitted after 1 January, 2021.

If you have not already done so, we invite you to upload the step-by-step protocols used in this manuscript to the Protocols Exchange, part of our on-line web resource, natureprotocols.com. If you complete the upload by the time you receive your manuscript proofs, we can insert links in your article that lead directly to the protocol details. Your protocol will be made freely available upon publication of your paper. By participating in natureprotocols.com, you are enabling researchers to more readily reproduce or adapt the methodology you use. [Natureprotocols.com](https://natureprotocols.com) is fully searchable, providing your protocols and paper with increased utility and visibility. Please submit your protocol to <https://protocolexchange.researchsquare.com/>. After entering your nature.com username and password you will need to enter your manuscript number (NG-A57213R2). Further information can be found at <https://www.nature.com/nature-portfolio/editorial-policies/reporting-standards#protocols>

Sincerely,
Wei

Wei Li, PhD
Senior Editor
Nature Genetics

New York, NY 10004, USA
www.nature.com/ng